# Biomedical Properties of Propolis on Diverse Chronic Diseases and Its Potential Applications and Health Benefits

**DOI:** 10.3390/nu13010078

**Published:** 2020-12-29

**Authors:** Nelly Rivera-Yañez, C. Rebeca Rivera-Yañez, Glustein Pozo-Molina, Claudia F. Méndez-Catalá, Adolfo R. Méndez-Cruz, Oscar Nieto-Yañez

**Affiliations:** 1Facultad de Estudios Superiores Iztacala, Universidad Nacional Autónoma de México, Tlalnepantla, Estado de México 54090, Mexico; nerivya@gmail.com (N.R.-Y.); rbkrivera14@gmail.com (C.R.R.-Y.); 2Laboratorio de Genética y Oncología Molecular, Laboratorio 5, Edificio A4, Facultad de Estudios Superiores Iztacala, Universidad Nacional Autónoma de México, Tlalnepantla, Estado de México 54090, Mexico; glustein@iztacala.unam.mx (G.P.-M.); mendezcatalacf@gmail.com (C.F.M.-C.); 3Laboratorio de Inmunología, Unidad de Morfofisiología y Función, Facultad de Estudios Superiores Iztacala, Universidad Nacional Autónoma de México, Tlalnepantla, Estado de México 54090, Mexico; renemen@gmail.com; 4Carrera de Médico Cirujano, Facultad de Estudios Superiores Iztacala, Universidad Nacional Autónoma de México, Tlalnepantla, Estado de México 54090, Mexico

**Keywords:** propolis, diabetes, obesity, cancer, chemical composition, bioactive compounds

## Abstract

The use of alternative medicine products has increased tremendously in recent decades and it is estimated that approximately 80% of patients globally depend on them for some part of their primary health care. Propolis is a beekeeping product widely used in alternative medicine. It is a natural resinous product that bees collect from various plants and mix with beeswax and salivary enzymes and comprises a complex mixture of compounds. Various biomedical properties of propolis have been studied and reported in infectious and non-infectious diseases. However, the pharmacological activity and chemical composition of propolis is highly variable depending on its geographical origin, so it is important to describe and study the biomedical properties of propolis from different geographic regions. A number of chronic diseases, such as diabetes, obesity, and cancer, are the leading causes of global mortality, generating significant economic losses in many countries. In this review, we focus on compiling relevant information about propolis research related to diabetes, obesity, and cancer. The study of propolis could generate both new and accessible alternatives for the treatment of various diseases and will help to effectively evaluate the safety of its use.

## 1. Introduction

### Definition and History

The term propolis is derived from two Greek words, pro (which means for or in defense of) and polis (which means the city); thus, propolis means in defense of the city or beehive [1]. Propolis is a natural, complex resinous mixture produced by honeybees by mixing products collected in tree buds, plants, saps, resins, and other plant sources [2], with beeswax and salivary enzymes (𝛽-glucosidase) [3]. Propolis is either a hard and breakable resin when it is cold, or soft, flexible, and very sticky when it is warm; it possesses different colorations, including brown, green, and red [3,4]. The bees use the propolis to cover surfaces, seal holes and close gaps, maintaining moisture and temperature stable in the hive throughout the year, and thus providing a sterile environment that protects them from microbes and spore-producing organisms, including fungi and molds [5]. In addition, bees use propolis as an embalming substance to mummify invaders, such as other insects, that have been killed and are too heavy to remove from the colony [5,6]. It is therefore considered to be a potent chemical defense against bacteria, viruses, and other pathogenic microorganisms that may invade the bee colony [7].

The ancient Greeks, Romans, and Egyptians were the first to use propolis, with applications in wound healing and as a disinfectant substance, and it is believed to have been in use since 300 BC [1,8,9]. Many other ancient civilizations, such as Chinese, Indian, and Arabian cultures, also used propolis to treat medical conditions such as skin lesions [10]. A large number of reports have been published showing numerous applications of propolis in treating various diseases due to its antibacterial, antifungal, antiparasitic, antiviral, antioxidant, anti-inflammatory, antitumor, and immunomodulatory properties, among others [11,12,13,14,15,16,17,18].

In the following sections we address some of the most recent research about the biomedical properties of propolis, in addition to the chemical compositions that have been identified in propolis from different parts of the world.

## 2. Chemical Composition of Propolis

To date, numerous studies have been published regarding the chemical composition and biological effects of propolis. Nevertheless, analysis of a large number of samples from different geographic regions has shown that its chemical composition is highly variable and difficult to establish because it depends on factors such as the vegetation and season at the collection site [6,19] and the bees species [20]. Propolis is generally composed of 50% resin, 30% wax, 10% essential oils, 5% pollen, and 5% other substances, which include minerals and organic compounds [6]. In addition, propolis samples from various regions of the world have been reported to contain over 300 different chemical compounds [21].

Using high throughput methods, such as mass spectroscopy, nuclear magnetic resonance, and gas chromatography coupled to mass spectrometry, it has been possible to identify several families of chemically active compounds. The main chemical groups present in propolis comprise phenolic acids or their esters, flavonoids, terpenes, aromatic aldehydes and alcohols, fatty acids, stilbenes, and 𝛽-steroids [21,22]. Additionally, propolis contains minerals such as magnesium, calcium, iodine, potassium, sodium, copper, zinc, manganese, and iron; vitamins such as B1, B2, B6, C, E, and D, in addition to provitamin A; a few fatty acids; and enzymes derived from bee glandular secretion or possibly pollen, such as succinic dehydrogenase, adenosine triphosphatase, glucose-6-phosphatase, acid phosphatase, 𝛼-amylase, 𝛽-amylase, 𝛼-lactamase, 𝛽-lactamase, maltase, esterase, and transhydrogenase. Polysaccharides such as starch and the di- and monosacharaides glucose, fructose, ribose, rhamnose, talose, gulose, and saccharose are also commonly present in propolis [23,24].

Flavonoids have been reported to be the main compounds of propolis, including flavones (luteolin), flavonols (quercetins and derivatives), flavanones (pinocembrin and naringenin), flavanonols (garbanzol and alnustinol), chalcones and dihydrochalcones, isoflavones (calycosin), isodihydroflavones (daidzein), flavans, isoflavans (vestitol and derivatives), and neoflavonoids (homopterocarpin and medicarpin) [25]. Terpenoids, which represent only 10% of the content, are responsible for the odor of propolis because they are volatile components of plants, and also contribute to its biological properties. Terpenoids include monoterpenes (terpineol, camphor), diterpenes (the main groups, such as ferruginol, junicedric acid and derivatives, pimaric acid, and totarolone), triterpenes (such as lupeol and derivatives, lanosterol, and amyrone and derivatives), and sesquiterpenes (such γ-elemene, valencene, 𝛼-ylangene, and 𝛼-bisabolol). Phenolic compounds include various acids such as cinnamic, *p*-coumaric, chicoric, caffeic, and fulric acids [25].

The general composition of propolis described above is known to depend on the collection area. It has been reported that propolis samples from Europe, North America, and other temperate zones are mainly composed of flavonoids (pinocembrin, pinobanksin, quercetin, chrysin, and galangin), and phenolic acids and their esters [26].

The American continent, due to its size, contains a wide diversity of ecosystems that provide very particular characteristics. Depending on their collection area, bees collect different elements to prepare propolis. Thus, it is common to observe different chemical compositions in various regions of the American continent (Table 1). An example is Canadian propolis, which is reported to contain compounds including benzoic acid, cinnamic acid, vanillinic acid, *p*-coumaric acid, ferulic acid, caffeic acid, palmitic acid, oleic acid, pinocembrin, pinobanksin, chrysin, galangin, isosakuranetin, alpinone, kaempferol, pinostrobin chalcone, and pinocembrin chalcone [27]. Similarly, the chemical composition of Mexican propolis has been studied by various research groups, identifying compounds such as rutin, naringenin, hesperetin, pinocembrin, caffeic acid phenethyl ester (CAPE), chrysin, naringin, kaempferol, quercetin, acacetin, luteolin, pinostrobin, izalpinin, rhamnetin, galangin, alpinetin, dillenetin, cinnamic acid, caffeic acid, ferulic acid, and syringic acid, and reporting differences in the content of compounds depending on the geographical area [28,29,30,31]. In Brazil, the composition of red, green, and brown propolis has been described, and different compounds have been reported, such as caffeic acid, gallic acid, transferulic acid, *p*-coumaric acid, catechin, epicatechin, kaempferol, formononetin, quercetin, rutin hydrate, luteolin, artepillin C, and CAPE [32,33,34,35]. As mentioned above, the geographical location directly determines the chemical composition of propolis; hence, in a study carried out with different samples of propolis from southern Brazil, three phenolic acids (gallic acid, caffeic acid, and coumaric acid), a prenylated derivative of cinnamic acid (artepillin C), and a flavonoid (pinocembrin) were identified [36]. Cases have also been reported in which, despite the geographical distance between two regions, the chemical composition of propolis has been found to be highly similar; for example, propolis from southern Nigeria is uncommon since it presents prenylated isoflavonoids, like Brazilian red propolis, and a high abundance of stilbenoid compounds [37]. Similarly, analysis of propolis samples from Venezuela revealed polyisoprenylated benzophenones in addition to the usual constituents found in samples of tropical regions [6,38,39]. Finally, analysis of Chilean propolis identified compounds such as apigenin, pinocembrin, quercetin, and CAPE [40].

Analysis of propolis from Africa showed that regions such as Kenya, Cameroon, Congo, and Ethiopia contains triterpenoids as its main chemical component [41,42,43]. Compounds such as ferulic acid, cis- and trans-caffeic acids, pinostrobin, and galangine have been identified in Egyptian propolis [44]. In a study by Touzani et al. in 2019 in the Moroccan region, nine phenolic compounds were identified with different concentrations, namely, caffeic acid, *p*-coumaric acid, ferulic acid, naringenin, pinocembrin, chrysin, galangin, pinobanksin, and quercetin [45]. Algeria propolis is composed of pinocembrin, chicoric and caffeic acids and their esters, galangin, diterpenic acids, flavonols such as chrysin, and aromatic acids [46,47].

Various researchers have shown interest in the study of European propolis. A large number of chemical compounds have been identified in Greek propolis, including protocatechuic acid, pinocembrin, kaempferol, apigenin, chrysin, galangin, chlorogenic acid, daidzein, ellagic acid, ferulicacid, gallicacid, hesperetin, hydroxytyrosol, luteolin, *p*-coumaric acid, pinobanksin, quercetin, tectochrysin, caffeic acid, sakuranetin, rhamnetin, CAPE, pinostrobin, syringic acid, kaempferide, acacetin, rutin, protocatechuic acid ethyl ester, resveratrol, phloridzin, maslinic acid, naringenin, eriodictyol, diosmetin, rosmarinic acid, myricetin, isorhamnetin, isosakuranetin, (+)-catechin, orientin, vitexin, trans-cinnamic acid, pinobanksin 3-O-acetate, cinnamylideneacetic acid, artepillin C, adipic acid, ursolic acid, suberic acid, genistein, hesperidin, corosolic acid, betulinic acid, isoferulic acid, naringin, tangeretin, diosmin, vanillin, and chrysoeriol [48]. Similarly, various compounds have been reported in Polish propolis, including aromatic acids, such as benzoic acid, dihydrocinnamic acid, cinnamic acid, *p*-hydroxybenzoic acid, vanillic acid, *p*-coumaric acid, *p*-methoxycinnamic acid, ferulic acid, dimethoxycinnamic acid, isoferulic acid, caffeic acid; fatty acids, such as palmitic acid, linoleic acid, oleic acid; flavonoids and chalcones, such as pinostrobin chalcone, pinocembrin chalcone, pinocembrin, pinobanksin chalcone, sakuranetin, pinobanksin, chrysin, galangin, isosakuranetin chalcone, isosakuranetin, alpinon chalcone, alpinon; and a number of esters [49]. Furthermore, in 2019 Pobiega et al. reported that five samples of Polish propolis from agricultural areas contained compounds such as pinocembrin, chrysin, pinobanksin, apigenin, kaempferol, quercetin, acacetin, galangin, (+/−)-pinostrobin, (+)-catechin, *p*-coumaric acid, ferulic acid, caffeic acid, vanillic acid, syringic acid, cichoric acid, and cinnamic acid [50].

Within the Asian continent, propolis samples have also been analyzed. In 2019, Jiang analyzed propolis from various areas of China, reporting compounds such as caffeic acid, *p*-coumaric acid, ferulic acid, iosuofmeraurlic acid, pinobanksin, naringenin, quercetin, kaempferol, apigenin, pinocembrin, chrysin, CAPE, and galangin [51]. In another study, the chemical compositions of propolis from China and the United States were determined. It was reported that the major compounds in the two propolis samples were similar, and included pinocembrin, pinobanksin-3-O-acetate, galanin, chrysin, pinobanksin, and pinobanksin-methyl ether; the study also identified other compounds, such as caffeic acid, *p*-coumaric acid, isoferulic acid, luteolin, quercetin, and kaempferol [53]. In propolis from Turkey, compounds such as gallic acid, (±)-catechin, caffeic acid, syringic acid, epigallokatechin, *p*-coumaric acid, trans-ferulic acid, trans-isoferulic acid, myricetin, trans-cinnamic acid, daidzein, luteolin, pinobanksin, (±)-naringenin, apigenin, kaempferol, chrysin, pinocembrin, galangin, and CAPE were identified [52].

It is known that the biological activity of propolis depends primarily on its chemical composition [7], bees collect tree buds, sap, and resin from different plant sources to produce various types of propolis; that is, the chemical composition of propolis is mainly dependent on the plant species in the area [51,53,54,55]. These variations in its composition directly influence its biomedical properties. Some reports have shown that propolis extract from the same country, for example Brazil or China, have variations in their antioxidant, antimicrobial, antiparasitic, cytotoxic and antitumoral properties, which is due to its large geographic extent and the considerable diversity of its flora [34,56,57]. These chemical variations determine the biomedical properties and in some cases the nutraceutical characteristics of each type of propolis. Although, for many researchers, this characteristic of propolis negatively affects its study and applications, we consider it to be a virtue, enabling the realization of a naturally different product with unique properties in each geographical region of each country, providing a source of new therapeutic alternatives due to the innumerable interactions, such as synergism and antagonism between the diversity of secondary metabolites that compose propolis [58,59,60]. For this reason, we consider that it important to continue with the study of propolis whose biological properties and chemical composition have not been determined.

Several abiotic factors such as light intensity, average temperature, humidity, wind speed, solar radiation, water availability, and rainfall, collectively defined as seasonal effects, can directly affect the concentration of plant components, including flavonoids and phenolic acids, which influence the chemical composition and bioactivity of propolis [61,62,63]. This aspect poses difficulties propolis classification and standardization, and the different solvents and extraction methods may influence its activity [32,64]. As such, universal standardization would be impossible; however, propolis biological properties should be linked to a detailed investigation of its chemical composition and its botanical sources [6,19]. A classification was proposed to group propolis into seven types: poplar, birch, green (alecrim), red (Clusia), Pacific and Canarian [6,19]. This classification it is proposed to group propolis into seven types: poplar, birch, green (alecrim), red (Clusia), Pacific, and Canarian [6]. Some works support this categorization, for example, showing that Macaranga-type Pacific propolis from different countries has very similar chemical composition and biomedical activities [63,65,66,67,68]. However, more research is needed to achieve reliable classification, specifying the characteristics of propolis from regions with arid or semi-arid climatic conditions of Africa and North America (like Mexico), where the vegetation includes a large number of endemic species. A limitation of this classification is that it only focuses on the identification of bioactive compounds that meet any of the seven proposed profiles and does not fully consider minor compounds or the synergistic and/or antagonistic effects they exert on biological activities. Standardization is urgently required for the correct application of propolis in the food and pharmaceutical industries in the near future [69,70]. Therefore, in this review, we address the different studies specifying the country of origin of each propolis sample. The interest in this complex natural product, which contains many active compounds, has increased due to its broad spectrum of biological and pharmacological properties [21,71,72], and because it is considered to be a promising natural source for the discovery of new pharmaceutical products to treat several types of diseases [7].

## 3. Biomedical Properties of Propolis on Diabetes, Obesity, and Cancer

Next, we analyze the main works that have been conducted using propolis from different countries to determine their beneficial effects on diseases such as diabetes, obesity, and cancer. Propolis has been extensively investigated because its constituents exhibit several properties that can be applied to treat these diseases. Most propolis presents bioactivities such as antioxidant and anti-inflammatory capacity, effects on the regulation of the cell cycle, and antitumor activity [48,56,73,74,75] due to the individual components of propolis, which are highly variable (Table 1) and are mainly grouped into phenolic acids or their esters, flavonoids, terpenes, aromatic aldehydes and alcohols, fatty acids, stilbenes, and 𝛽-steroids [21,22]. In vitro methods were reported to be useful for preliminary investigation of the potential of a natural product such as propolis; from these findings, the decision can be made to perform in vitro and clinical investigations [76]. Therefore, at the beginning of each section, the in vitro studies are described, followed by analysis of the in vivo and clinical studies will be analyzed, which provides more complete evidence of the effects of propolis in diseases.

A large number of studies of propolis from different countries have generated vast amounts of evidence of the beneficial effects of this bee product. Several reports particularly emphasized on the various activities of Macaranga-type Pacific propolis, such as antioxidant, anti-inflammation, anticancer, antidiabetic, and longevity-extending effects; besides this, its chemical composition is one of the best described [63,66,67,77,78]. In this review, we compile studies of with propolis of different origins and we integrate the studies with Macaranga-type propolis that refer to the country of origin.

### 3.1. Effect of Propolis on Diabetes

Diabetes mellitus is a heterogeneous and chronic disease, characterized by hyperglycemia caused by an absolute or relative deficiency in insulin secretion or action, resulting in impaired function in carbohydrate, lipid, and protein metabolism [79,80]. Dyslipidemia is also a common feature of diabetes, and is characterized by elevated triglyceride and low-density lipoprotein cholesterol (LDL-C) levels [81]. Hyperglycemia and dyslipidemia easily induce extensive oxidative stress, which causes serious cellular dysfunction [82,83]. In addition, persistent hyperglycemia increases the production of reactive oxygen species (ROS) in several tissues [84] and also diminishes biological antioxidative mechanisms by glycation of the scavenging enzymes [85]. Thus, effective control of hyperglycemia in diabetic patients is critical for reducing the risk of micro- and macrovascular complications [86]. In this context, the prevalence of diagnosed diabetes is dramatically increasing and has become a major concern globally [87]. According to Saeedi et al. the International Diabetes Federation reported that the prevalence of diabetes was estimated to be 463 million in 2019, and is projected to reach 463 million by 2030 and 700 million by 2045 [88]. Thus, alternative therapies based on natural sources with antihyperglycemic properties, such as propolis, could play an important role in the management of diabetes, delaying the development of diabetic complications, and correcting metabolic abnormalities.

Numerous studies have shown that treatment with propolis decreases the glucose levels in the blood (Table 2). Alloxan-induced hyperglycemia in Wistar rats treated with Nigerian propolis (300 mg/kg) orally for 28 days decreased fasting blood glucose level in a manner similar to that in a group treated with metformin. Furthermore, the serum malondialdehyde (MDA) level was reduced and superoxide dismutase (SOD) levels were elevated in the propolis group. Histologically, improvement was evident in the pancreas and liver of the group treated with propolis. These findings suggest that Nigerian propolis confers protection against hyperglycemia-induced oxidative stress in both the liver and pancreas of Wistar rats [89]. Similarly, other investigations have reported that oral administration of Nigerian propolis (200 mg of propolis/mL of Tween 80) for 42 days in diabetic rats induced by alloxan, decreases the fasting blood glucose level at two weeks, and partially decreases glycated hemoglobin A1c (HbA1c) at six weeks of treatment. Moreover, Nigerian propolis treatment increased serum high-density lipoprotein (HDL), and reduced serum very low-density lipoprotein (VLDL) and cholesterol levels. The study concluded that Nigerian propolis contains compounds exhibiting hypoglycemic, antihyperlipidemic, and HbA1c-reducing activities [90]. Although studies on diabetes animals models induced with alloxan are not the most reliable in terms of the similarity of the damage process of this disease, they provide initial insights into whether propolis shows any activity on diabetes; however, it is necessary corroborate the results with a model that better resembles the course of diabetes, such as streptozotocin-induced animal models. We agree Jain et al. (2011) and Misra et al. (2012), who reported that alloxan shows doubtful potential as a drug regarding the induction of experimental diabetes. This, however, is contradictory to many studies where alloxan has been successfully used [91,92]. Another limitation of these works is the absence of the chemical composition; as we have already mentioned, this is essential for the correct pharmaceutical and medical application of propolis [70].

The activity of propolis has also been tested in animals with type 1 diabetes. In streptozotocin (STZ)-induced type I diabetic mice, treatment with Saudi Arabian propolis (100 mg/kg body weight) for one month improved body weight, decreased blood glucose levels, and restored serum insulin; it also restored the plasma cytokine (IL-1β, IL-6 and TNF-α) and reactive oxygen species (ROS) levels, and the lipid profile, to nearly normal levels [94]. In another study, treatment for four weeks with propolis from Malaysia (300 mg/kg/day propolis orally) reduced the fasting blood glucose in STZ-induced type 1 diabetic Sprague Dawley (SD) rats [95]. Similarly, treatment with Iranian propolis (100 and 200 mg/kg) for six weeks, in type 1 diabetes rats (Wistar) induced by STZ, inhibited bodyweight loss and reduced serum glucose levels. Furthermore, propolis reduced the MDA content and increased the activity of SOD and glutathione peroxidase (GPx), in addition to the total antioxidant activity in the kidney tissue of diabetic rats. The study concluded that Iranian propolis can enhance the antioxidant levels and histopathological changes in the kidneys of rats, and that most of the favorable effects of propolis are mediated by a reduction in blood glucose levels in diabetic animals [97]. In a model of STZ-induced type 1 diabetes in SD rats, the oral administration of Chinese and Brazilian propolis (a group with 10 mg of Chinese propolis per 100 g of body weight and another group with 10 mg of Brazilian propolis per 100 g of body weight) for eight weeks showed that both types of propolis inhibited body weight loss and decreased blood glucose level. HbA1c levels were also reduced in the Chinese propolis-treated group, whereas Brazilian propolis showed a trend toward reduced levels. Moreover, Brazilian propolis (but not Chinese propolis) slightly decreased total cholesterol levels, and neither propolis influenced blood triglyceride, low-density lipoprotein cholesterol (LDL-C), or high-density lipoprotein cholesterol (HDL-C) levels. Moreover, oxidative stress in blood, liver, and kidney was improved to various degrees by both types of propolis. Similarly, apparent reductions in the levels of alanine transaminase, aspartate transaminase, blood urea nitrogen, and urine microalbuminuria-excretion rate demonstrated the beneficial effects of both types of propolis on hepatorenal functions, suggesting that Chinese and Brazilian propolis can alleviate symptoms of diabetes mellitus in rats and that these effects may partially be due to their antioxidant ability [102]. The possible antidiabetic, hypolipidemic, and antioxidant effects of Brazilian green propolis have also been investigated in type I diabetes STZ-induced (Wistar) rats. Daily treatment (200 mg/kg) for five weeks resulted in amelioration of the animals’ body weight loss, reduction in serum glucose, triglycerides, total cholesterol, and LDL-C, and an increase in serum HDL-C compared to the normal control group. In addition, there was a reduction in pancreatic lipid peroxides, such as MDA and serum nitric oxide (NO), and a marked increase in serum reduced glutathione (GSH), catalase (CAT), and pancreatic SOD in propolis-treated rats. The study findings suggest that propolis offers promising antidiabetic and hypolipidemic effects that may be attributed to its potent antioxidant potential [103]. Research on the effect of propolis in type I diabetes has reported different activities such as reduction of glucose and HbA1c levels, improvement of the lipid profile, restoration of antioxidant capacity, and an inhibition of body weight loss. The results are consistent and suggest an antidiabetic effect; however, many authors agree, as do we, that it is necessary to conduct studies that evaluate more pharmacological and biochemical effects of propolis, focused on the search for new, better, and more efficient complementary therapies for the treatment of diabetes [106,107].

Similarly, the effect of propolis has been studied in animal models with type 2 diabetes. Treatment with Indonesian propolis (each group with doses of 50, 100, and 200 mg/kg respectively) for 15 days in diabetic type 2 mice (BALB/c) induced with S961 peptide lowered blood glucose level in a dose-dependent manner [98]. Another study investigated the effect of the treatment of Brazilian propolis (100 and 300 mg/kg) for four weeks on insulin resistance in Otsuka Long-Evans Tokushima Fatty (OLETF) rats, a non-insulin-dependent type 2 diabetic model. Propolis treatment decreased the plasma levels of insulin and the insulin resistance index (homeostasis model assessment-insulin resistance, HOMA-IR), without affecting blood glucose levels, suggesting that Brazilian propolis could be an effective and functional food to prevent development of insulin resistance [104]. It has also been reported that an eight weeks diet containing Brazilian propolis (groups at 0.1% and 0.05% respectively) in OLETF rats decreased blood glucose associated with a reduction in plasma insulin levels, although plasma free fatty acids (FFAs) and β-hydroxy butyrate were not altered. These results suggest that dietary propolis improves blood glucose and insulin sensitivity [105].

In 2019, El Menyiy et al. studied the effect of Moroccan propolis in STZ-induced type 2 diabetic rats (Wistar). At the end of treatment (concentrations of 50 and 100 mg/kg), propolis caused a lowering of blood glucose and lactic acid dehydrogenase, increased body weight, and ameliorated dyslipidemia and abnormal liver and kidney function caused by diabetes. The effect of propolis was dose dependent; in a high dose it was more potent than glibenclamide, indicating that propolis from Morocco exhibited strong antihyperglycemic, antihyperlipidemic, and hepato-renal protective effects in diabetes [93]. Similarly, treatment with Saudi Arabia propolis (20% *w/w*) for four weeks in STZ-induced type 2 diabetic rats reduced glucose levels and lipid peroxide, accompanied by an increase in SOD, CAT, and glutathione-S-transferase enzyme activities. The treatment also reduced carboxymethyl lysine, IL-6, and immunoglobulins to levels similar to those of the control [79].

Propolis from Malaysia has been reported to have antihyperglycemic, antioxidant, and anti-inflammatory potential in STZ-induced type 2 diabetic SD rats. After four weeks, groups treated with propolis alone and propolis (300 mg/kg/day) combined with metformin (300 mg/kg/day) showed reductions in fasting blood glucose and serum insulin level, and body weight increased compared to the diabetic group. Furthermore, pancreatic antioxidant enzymes, total antioxidant capacity, IL-10, and proliferating cell nuclear antigen (PCNA) increased, whereas MDA, nuclear factor-kappa B (NF-kB), tumor necrosis factor alpha (TNF-α), IL-1β, and cleaved caspase-3 decreased in the groups treated with propolis alone and propolis combined with metformin compared with the diabetic group. In addition, histopathology of the pancreas showed an increased islet area and number of beta cells in both groups. Moreover, propolis from Malaysia showed in vitro α-glucosidase inhibition activity [96]. Similarly, it has been reported that oral administration of encapsulated Chinese propolis (at a concentration of 50, 100, and 200 mg/kg/day) for 10 weeks in type 2 diabetic SD rats induced by a high-fat-diet (HFD) and low-dose STZ attenuated the fasting blood glucose and triglycerides, and improved insulin action. However, there were no effects on body weight, total cholesterol, HDL-C, and LDL-C, indicating that encapsulated Chinese propolis can control blood glucose, modulate lipid metabolism, and improve insulin sensitivity in type 2 diabetic rats [101]. Similarly, in a type 2 diabetes SD STZ/HFD-induced rat model, the administration of Taiwanese propolis (183.9 mg/kg/day and 919.5 mg/kg/day) for eight weeks delayed the development and progression of diabetes and reduced the severity of β-cell failure. Treatment also attenuated inflammation (serum TNF-α, IL-1β, and IL-6) and ROS. Moreover, propolis reduced the MDA content, and increased the activity of SOD and GPx and the total antioxidant activity in the kidney tissue of diabetic rats. Taiwanese propolis also promoted hepatic genes *PPAR-α* and *CYP7A1*, which are related to lipid catabolism and removal, indicating that it may delay the progression of diabetes through anti-inflammation and anti-oxidation effects, and balancing of lipid metabolism [99]. As mentioned in all previous research studying the effect of propolis in animal models of type 2 diabetes, this natural product has a wide variety of antidiabetic properties, since it can control blood glucose, modulate lipid metabolism, improve insulin sensitivity, increase the activity of antioxidant enzymes in different tissues, an increase in the area of the islets and the number of beta cells is even mentioned by some authors; however, the conclusions provided by the authors are limited, so different studies needs to be undertaken that focus on elucidating the mechanisms of action through which propolis affects diabetes [108,109].

The antidiabetic effect of propolis from Korea has been described through inhibition of gluconeogenesis, based on analysis of the expression and enzyme activity of glucose-6-phosphatase (G6Pase) in hepatocytes. Treatment with Korean propolis (different concentrations from 0 to 50 µg/mL) reduced the expression and enzyme activity of G6Pase in HepG2 cells. It also inhibited serine and tyrosine phosphorylation, thus inhibiting the activity of GSK3α and β, which maintain G6Pase gene expression. Therefore, the findings suggest that propolis from Korea may be a potential antidiabetic agent for the treatment of insulin-insensitive diabetes [100].

Similarly, flavonoids and other compounds in propolis have been found to possess antidiabetic activity (Table 3) [110,111]. It was shown that Chinese propolis administered (100 and 200 mg/kg/day) for 12 weeks in STZ-induced type 1 diabetic rats was able to attenuate diabetes, decreasing the levels of fasting blood glucose and HbA1c, which also resulted in the reduction in MDA, ROS, and reactive nitrogen species (RNS). Furthermore, different compounds, such as caffeic acid, *p*-coumaric acid, ferulic acid, isoferulic acid, pinobanksin, quercetin, kaempferol, apigenin, pinocembrin, chrysin, CAPE, and galangin, were identified [112]. In 2018, Rivera-Yañez et al. reported that oral administration of Mexican propolis (0.3 g/kg/day) for 15 days decreased the blood glucose and the loss of body weight, and increased plasma insulin levels in STZ-induced type 1 diabetic mice. Furthermore, increased activities of the enzymes SOD, CAT, and GPx in serum were observed in diabetic mice treated with propolis. In addition, compounds such as pinocembrin, quercetin, naringin, naringenin, kaempferol, acacetin, luteolin, and chrysin were identified, demonstrating that Mexican propolis possesses hypoglycemic and antioxidant activities, and can alleviate symptoms of diabetes mellitus in mice. Moreover, these effects may be directly related to the chemical composition of propolis, because it possesses different bioactive compounds [29]. We consider that these effects are due to what was reported by some authors that have administered flavonoids, such as quercetin, naringin, or genistein, and have informed reduced blood glucose concentrations and detected insulin in serum or islets, these effects likely resulted from changes in Ca2+ metabolism, thus facilitating the hypoglycemic effects of flavonoids [113,114,115,116]. Moreover, different studies have reported that the administration of some flavonoids, such as acacetin, luteolin, chrysin, kaempferol, or naringenin, decreases glucose levels in diabetic rats and mice, and inhibitory effects against the enzymatic activity of 𝛼-glucosidase [117,118,119,120]. These results are promising, so in the future, clinical studies administering propolis or its main bioactive compounds are required. Some clinical studies have already focused on the application of compounds such as quercetin in the treatment of diabetes [121]. Another study investigated whether Brazilian propolis (50 or 250 mg/kg body weight) affects glucose uptake and translocation of insulin-sensitive glucose transporter (GLUT) 4 in skeletal muscle cells. In L6 myotubes, the propolis promoted GLUT-4 translocation and glucose uptake activity. Regarding the mechanism of GLUT-4 translocation, propolis induced both phosphatidylinositol 3-kinase (PI3K) and 5′-adenosine monophosphate-activated protein kinase (AMPK) phosphorylation in a dose-dependent manner in L6 myotubes. The main polyphenols found in the propolis extract, artepillin C, coumaric acid, and kaempferide, promoted GLUT-4 translocation in L6 myotubes. Additionally, these compounds activated both PI3K- and AMPK-dependent dual-signaling pathways. However, only kaempferide increased glucose uptake activity. Moreover, single oral administrations of propolis lowered postprandial blood glucose levels in mice. It was also confirmed that propolis promoted phosphorylation of both PI3K and AMPK in rat skeletal muscle, indicating that Brazilian propolis has the potential to prevent hyperglycemia through the promotion of GLUT-4 translocation in skeletal muscle and that kaempferide is a potential active compound in propolis [110]. As mentioned above, many studies have evaluated the effect of propolis and some of its bioactive components on diabetes; although this is promising, it is necessary to evaluate these possible mechanisms of action in in vivo models [122], since they have to consider other factors such as oxidative stress, chronic low-grade inflammation, and the response of each individual to treatment, which would help to better understand the disease and its possible complementary treatments [122,123].

However, although a wide variety of studies exist that demonstrate the antidiabetic properties of propolis, as described above, few clinical studies have reported the efficacy of propolis on diabetes mellitus (Table 4). In this context, the effect of Iranian propolis on glucose metabolism, insulin resistance, and renal and liver function, in addition to inflammatory biomarkers in patients with type 2 diabetes mellitus, was evaluated. Treatment with Iranian propolis capsules (500 mg per capsule) twice daily for 90 days improved the serum levels of HbA1c, two-hour postprandial glucose, blood insulin, and insulin resistance indices (including HOMA-IR and Homeostasis Model Assessment of β-cell function, HOMA-β) in comparison with the placebo group. Moreover, blood C-reactive protein and TNF-α levels were reduced following propolis treatment. A notable reduction in serum liver transaminases (ALT and AST) and blood urea nitrogen concentrations in the propolis group was also observed, indicating that Iranian propolis treatment can mitigate systemic inflammation and be a useful treatment for preventing liver and renal dysfunction in patients with type 2 diabetes mellitus [124]. In a similar study, type 2 diabetic patients took Iranian propolis capsules 3 times per day (500 mg per capsule) for eight weeks. At the end of the study, the levels of fructosamine and oxidized low-density lipoprotein (LDL) decreased, and CAT activity improved, indicating that an 8 week intake of propolis as a supplement in type II diabetic patients can improve antioxidant defense and reduce production of hyperglycemia-induced products such as fructosamine [125]. Another study assessed the effect of oral capsules containing Iranian propolis (three capsules/day for eight weeks; 500 mg per capsule) on blood glucose, insulin resistance, and antioxidant status in type 2 diabetes. After two months, fasting blood sugar, two-hour postprandial glucose, blood insulin, insulin resistance index (HOMA-IR), and HbA1c were reduced in patients treated with propolis. Additionally, intake of propolis significantly increased the total antioxidant capacity and activity of GPx and SOD, suggesting that propolis treatment can be helpful as a diet complement in patients with type 2 diabetes via improvement in glycemic status, reduction in insulin resistance, and amelioration in antioxidant status [126]. Similarly, another study determined the effect of administration of a propolis-containing pill (3 pills/day for 12 weeks; 300 mg per capsule) produced by an Iranian company on glycemic control, serum lipid profile, and insulin resistance in patients with type 2 diabetes. Propolis reduced fasting blood glucose, HbA1c, cholesterol total, and LDL, whereas it failed to affect blood triglyceride level, HDL, VLDL, and insulin resistance indices (including HOMA-IR and HOMA-β) in diabetic patients, indicating that the daily intake of the propolis supplement results in improvement of glycemia and some serum lipid levels in patients with type 2 diabetes [127].

The administration of Chinese propolis capsules for 18 weeks (900 mg daily) in patients with type 2 diabetes mellitus has also been evaluated. At the end of the study, no difference was found between the Chinese propolis group and the control group in the different parameters analyzed. However, serum GSH increased, accompanied by a reduction in lactate dehydrogenase (LDH) in the Chinese propolis group [128]. In a similar study, treatment with capsules of Brazilian green propolis (900 mg daily) for 18 weeks increased GSH in the blood and decreased LDH and TNF-α [129]. Therefore, like Chinese propolis, Brazilian green propolis appears to exhibit mild antidiabetic effects via antioxidant activity [128,129]. However, it is important to continue clinical studies to verify whether different doses or longer administration of propolis could be beneficial in the prevention and care of diabetes. Although few clinical studies have mentioned some effects of propolis on diabetes, most of the research is with Iranian propolis. As such, clinical studies on propolis from different regions should be conducted, since it is important to consider the variations in the biomedical properties of different propolis. In addition, the most important omission in these clinical studies is that they did not report the chemical composition of propolis, nor did mention whether there is a history of in vitro and in vivo studies that support whether these propolis types have any type of biological activity or how it was decided to conduct clinical studies [128,129].

Overall, the study of natural products, such as propolis and some of its pure compounds, could provide potential strategies for the development of effective and safe drugs for alternative treatment of diabetes, lowering blood glucose and enhancing the antioxidant system and insulin action and secretion. However, more studies are needed to elucidate the molecular mechanisms involved in glucose and lipid metabolism in diabetes, and to identify additional therapeutic benefits.

### 3.2. Effect of Propolis on Obesity

Obesity is a significant public health threat because it is associated with increased incidences of cancer, type 2 diabetes, dyslipidemia, cardiovascular diseases, among other pathologies [130,131]. The global prevalence of obesity has reached epidemic proportions, with an estimated 671 million people suffering from obesity in 2016 [132]. Furthermore, obesity is the fifth leading cause of death globally [130]. A long-term high-fat or unbalanced diet contributes to obesity [133,134]. To increase adipose tissue mass, adipocytes undergo hypertrophy (an increase in size) and hyperplasia (an increase in number). Large lipid-laden adipocytes in hypertrophic tissue secrete several mediators that trigger metabolic dysfunction [135]. In parallel with adipocyte hypertrophy, the infiltration of immune cells, such as macrophages, into adipose tissue is frequent, causing mild inflammation in the adipose tissue [136,137] and increasing circulating levels of adipokines, fatty acid mediators (lipokines), and miRNA-containing exosomes, which affect energy metabolism in the liver and skeletal muscle [138,139,140]. In addition, chronically obese individuals acquire leptin resistance and show deterioration of energy metabolism [141]. Thus, the search for natural alternatives, such as propolis, that can help in the treatment of obesity for the management of weight gain, lipid metabolism regulation, and consequent pathological complications induced by obesity, is now of utmost importance.

Although little research exists relating to the effect of propolis on obesity (Table 5), Brazilian red propolis treatment (0–100 µg/mL) for three days has been found to induce adiponectin mRNA in 3T3-L1 preadipocytes, probably through activation of the adiponectin promoter by peroxisome proliferator-activated receptor γ (PPAR-γ). In addition, in the same study, propolis treatment for eight days restored adiponectin expression in TNF-α-treated, differentiated 3T3-L1 cells, suggesting the value of Brazilian red propolis as a diet supplement for prevention and treatment of obesity and obesity-associated disorders [142]. Similarly, an in vitro assay using differentiated 3T3-L1 adipocytes showed that Brazilian green propolis (100 µg/mL) directly elevated leptin expression [143]. Although in vitro studies have helped provide an overview of the activity of propolis on obesity, most only evaluate only a chemical or molecular aspect, so its effect is still partial. The studies did not consider that this disease is multifactorial; therefore, in vivo studies are essential to reach clinical trials [144,145].

One study showed that intraperitoneal injection of Brazilian green propolis twice per week for 12 weeks caused feeding suppression in obese C57BL/6 mice induced with HFD, and increased leptin mRNA production in the visceral adipose tissues. These findings suggest that Brazilian propolis causes leptin induction in adipocytes and has the potential to attenuate feeding in obese mice and possibly prevent obesity [143]. Similarly, the anti-adiposity effects of Brazilian green propolis have been observed using C57BL/6JHamSlc-ob/ob mice (a monogenic mutant obese model). An intraperitoneal injection of propolis (100 mg/kg) twice per week for 12 weeks reduced mesenteric adipose tissue mass, whereas weights of epididymal and inguinal adipose tissue were not modulated. Plasma cholesterol levels were also attenuated, but triglyceride and non-esterified free fatty acids levels were not affected. The results indicate that Brazilian green propolis has the potential to normalize dyslipidemia, although effects differed between the weights of adipose tissues [146]. Another study examined the hypolipidemic effect of Brazilian propolis in a C57BL/6N mice obesity model induced by an HFD. The mice administrated with propolis intragastrically (doses of 5 mg/kg or 50 mg/kg) twice daily for 10 days displayed reductions in body weight gain, weight of visceral adipose tissue, liver and serum triglycerides, cholesterol, and non-esterified fatty acids. Real-time PCR analysis of the liver also showed down-regulation of mRNA expression associated with fatty acid biosynthesis, including fatty acid synthase, acetyl-CoA carboxylase a, and sterol regulatory element binding protein in the propolis-administrated mice. In the same study, obese C57BL/6N mice (induced by an HFD) were given propolis for four weeks and showed reductions in weight gain, serum non-esterified fatty acids, and lipid accumulation in the liver. The results suggest that Brazilian propolis prevents and mitigates high-fat diet-induced hyperlipidemia by down-regulating the expression of genes associated with lipid metabolism [147]. Other research has demonstrated that, in C57BL/6 mice induced by an HFD, a supplemented Brazilian propolis diet (0.2% crude propolis *w/w*) of two or five weeks reduced blood triglyceride levels, although it failed to reduce blood total cholesterol and HDL levels [148]. In 2020, Zheng et al. showed the beneficial effects of dietary Chinese propolis (150 and 300 mg/kg by oral gavage for nine weeks) on weight in HFD-fed C57BL/6 mice. Dietary propolis supplementation reduced body weight and improved insulin resistance in the HFD-fed mice in a dose-dependent manner. Propolis treatment also reversed liver weight loss and triglyceride accumulation in association with hepatic steatosis, suggesting that Chinese propolis can prevent diet-induced obesity, and indicating the possibility of using propolis as an alternative method for weight control [150].

Brazilian propolis has been found to affect body fat and lipid metabolism. HFD-induced obese Wistar rats were fed propolis (at two concentrations of propolis 0.05 and 0.5% *w/w*) for eight weeks. At the end of treatment, the supplemented propolis diet repressed the weight gain of mesenteric, perirenal, and total white adipose tissues; the level of the protein PPAR-γ in the adipose tissues was lower. In plasma and liver, the propolis group showed a reduced level of cholesterol and triglyceride. The liver 3-hydroxy-3-methylglutaryl coenzyme A (HMG-CoA) reductase protein was also lower, and the hepatic PPAR-α protein level higher, in the propolis group. In another test, after feeding the mice with olive oil, the serum triglyceride level was lower in the propolis-treated group. The results indicate that it is possible that the administration of Brazilian propolis improves the accumulation of body fat and dyslipidemia via the change of the expression of proteins involved in adipose depots and lipid metabolism [149]. In the in vivo studies mentioned in this review, the different effects of propolis on obesity have been reported through various mechanisms suppressing feeding in mice, inducing leptin in adipocytes, reducing adipose tissue mass, and improving dyslipidemia. However, obesity encompasses a large number of factors that need to be considered for its study, which continues to be a challenge in the field of research, which explains the scarcity in studies. To determine an optimal effect of propolis on this disease, a large investigation that can combine aspects related to obesity, such as diabetes, metabolic syndrome, and hypertension, is required to reach a reliable conclusion regarding the activity of propolis on obesity. Clinical studies are also required for this complex disease [151].

In addition, research has demonstrated the potential health benefits of natural flavonoids against obesity [152], and some of these compounds have also been identified in propolis. The effect of oral administration for 30 days of propolis from Croatia (50 mg/kg) on the lipid profile and the hepatorenal function in C57BL/6N mice fed with HFD was examined. The propolis diet reduced the serum triglycerides, total cholesterol, and LDL levels, whereas the level of HDL was not modified. The modified diet also reduced the weight of mice and lipid accumulation in the liver and showed significant in vitro antioxidative activities. Compounds such as quercetin, naringenin, caffeic acid, galangin, and chrysin were identified. The results indicate that Croatian propolis, as a mixture of polyphenolic compounds, possesses antioxidative properties and has positive effects on hepatorenal functions and lipoprotein distribution, thus suggesting this propolis may have potential as treatment for obese patients with hyperlipidemia [153]. Existing different studies have reported that in growing preadipocytes, quercetin extensively decreases the expression of LPL, sterol regulatory element-binding protein 1c (SREBP1c), and PPAR-γ, a key adipogenic transcription factor [154,155]. Quercetin also caused dose- and time-dependent increases in lipolysis in rat adipocytes [156]. The anti-obesity effect of naringenin is through the reduction in adipose tissue mass and inhibition of preadipocyte proliferation [157]. Moreover, this compound also remarkably increased the activity of various enzymes required for fatty acid oxidation in hepatocytes, such as acetyl-coenzyme A acetyltransferases (ACAT, also known as thiolase), acyl-coenzyme A oxidase, carnitine O-octanoyl transferase (COT, also known as medium-chain/long-chain carnitine acyltransferase), and 3-ketoacyl-coenzyme A [158].

As previously mentioned, it is important to identify alternatives for obesity treatment. Natural products, such as propolis and some of its compounds, could offer significant therapeutic value for help in the control of fat accumulation and weight gain. Propolis could also have potential as a functional ingredient in efficient and safe drugs for the prevention and treatment of obesity and complications induced by this pathology.

### 3.3. Effect of Propolis on Cancer

Cancer is characterized by aberrant cell cycle activity. This occurs either as a result of mutations in upstream signaling pathways or by genetic lesions within genes encoding cell cycle proteins [159]. Cancer is an important cause of morbidity and mortality globally, with variation across countries and states. Among males, lung cancer is the leading cause of cancer death, followed by liver and stomach cancer. Among females, breast cancer is the leading cause of cancer death, followed by lung, colorectal, and cervical cancer [160]. In addition to current cancer treatments, such as surgery and chemotherapy, combination therapies exist. However, in several cases, these therapies have led to a significant increase in adverse effects and may thus not be tolerated by many patients [159]. Thus, the innumerable applications of natural products offer considerable opportunities for the improvement of human well-being and health. As such, propolis and some of its active compounds could have potential in the development of drugs that can act as complementary therapeutic agents in cancer treatments.

Antitumoral effects of propolis on cancer cells have been described in a number of studies (Table 6). Diverse research has reported that propolis from distinct countries (Brazil 50 μg/mL; Mexico 15–500 μg/mL; and Thailand 0–1 mg/mL) presents antitumor activity in different tumor cell lines (human ovarian cancer cells, OVCAR-8; human colon cancer cells, HCT-116; human promyelocytic leukemia cells, HL-60; human glioblastoma cells, SF-295; human cervical cancer cells, HeLa, SiHa, and CaSki; human adenocarcinomic alveolar basal epithelial cells, A549) [31,56,161]. Notably, that these studies report the evaluation of various biomedical activities of different extracts of propolis, so the reported evidence is limited to a cytotoxic approach; therefore, we consider is necessary to include controls with non-malignant cells to determine if the effect is selective for cancer cells. However, only one of the reports described the chemical composition, providing a basis for discussion. The description of the chemical composition is relevant for some authors since they mention that the systematic classification of propolis will be impossible without it [6].

In Europe, research has also been undertaken on the antitumor effects of propolis. In 2020, Wezgowiec and collaborators reported that propolis from Poland possesses antitumor activity against tongue cancer cells (human squamous cell carcinoma derived from tongue, SCC-25), at concentrations of 100, 500, and 1000 µg/mL, affecting cell viability and cellular proliferation after 24 h incubation, but does not have an effect on gingival fibroblasts, indicating the potential of Polish propolis as a natural product with cancer-selective antiproliferative activity [162]. Similarly, research has examined the antitumor effect of propolis from northeastern and central Portugal (0–100 µg/mL) on normal and cancerous renal cells derived from human renal cell carcinoma patients, and in the A-498 (Human renal epithelial cancer) cell line. Propolis exhibited selective antiproliferative activity against malignant cells compared to normal cells. In vitro, human renal carcinoma cell growth was strongly inhibited by propolis in a concentration-dependent manner. In the A-498 cell line, propolis from northeastern Portugal also exerted a higher antitumor effect than propolis from central Portugal. The study conclusion was that Portuguese propolis could be a source of antitumor agents [163]. Another study investigated the in vitro antitumor activity of Portuguese propolis on the human colon carcinoma cell line HCT-15. Propolis exhibited an antitumor effect against HCT-15 cells (5–50 µg/mL), both in terms of inhibition of viability and cell death, in a dose- and time-dependent manner. It was found that the antitumor effect involves disturbance in tumor cell glycolytic metabolism, evidenced as a decrease in glucose consumption and lactate production. The study results suggest that Portuguese propolis can be a potential therapeutic agent against human colorectal cancer [164]. Other research demonstrated the antitumor and antiangiogenic effects of propolis from Portugal (using concentrations of 5–100 µg/mL) on different tumor cells. Propolis decreased cell viability of different tumor cells (MDA-MB-468 and MCF7 (breast cancer), 22RV1 (prostate cancer), U251, and SW1088 (glioblastoma)), of which MDA-MB-231 (breast cancer) and DU145 (prostate cancer) were the most sensitive. This effect was not present on non-tumor cells (non-tumor breast cell line HB4a and the prostate epithelial cell line PNT-1) and fibroblasts (human dermis fibroblasts hDFb190). Portuguese propolis also decreased MDA-MB-231 and DU145 cell proliferation and migration, with cell cycle changes, and increased cell death. The increase observed in glucose consumption and lactate production could be explained in MDA-MB-231 cells by an increased expression of different metabolism-related proteins. The propolis induced a decrease in HBMEC (human brain microvascular endothelial cells) total biomass and proliferation, and decreased vessel sprouting in the chicken chorioallantoic membrane. The study concluded that Portuguese propolis could be a good candidate for cancer drug development because it affects different characteristics that dictate tumorigenesis [165]. Although in vivo tests were not performed, they provide relevant results since the antitumor, antiproliferative, and antiangiogenic effects of several European propolis extracts were reported. So, the next step is to conduct more in vivo studies to consolidate the use of propolis and its components to enable the development of clinical studies. Rodent models of cancer have revolutionized our ability to study gene and protein functions in vivo and better understand their molecular pathways and mechanisms [173].

Similarly, in Asia, the antitumor properties of Iraqi propolis on HL-60 and HCT-116 cell lines have been evaluated at concentrations of 5, 15, and 25 µg/mL. Propolis induced an antitumor effect against HL-60 cells and an inhibitory effect in the colony potential of HCT-116 cells. Moreover, the apoptosis in HL-60 cells was associated with down-regulation of Bcl-2 and the activation of Bax, whereas in HCT-116 cells necrotic features were observed. The size of cells was also dramatically increased by swelling of cytoplasm and loss of membrane integrity, cell rupture, and release of cellular contents; similarly, the propolis induced cell cycle perturbations in both cell lines. The exposure of HL-60 cells to propolis caused an increase in the levels of γ-H2AX in a dose-dependent manner and was associated with induction of apoptosis, indicating that Iraqi propolis could be a promising antitumor agent [166]. The antitumor effects of propolis from Korea on HL-60 cells (0–50 μg/mL) were also investigated, with dose- and time-dependent decreases observed in the proliferation of HL-60 cells. The propolis also induced both the activation of caspase-3 and cleavage of poly (ADP-ribose) polymerase, and the release of cytochrome c from mitochondria to cytosol, demonstrating that the antitumor effect of propolis on HL-60 cells is caused by inducing apoptosis through the mitochondrial pathway [167]. Other studies have examined the antitumor effects of Chinese propolis (0–100 μg/mL) in four human colon carcinoma cell lines, CaCo2, HCT-116, HT29, and SW480. The propolis showed marked dose-dependent antitumoral activity in HCT-116, HT29, and SW480 cell lines. In the HCT-116 cell line, Chinese propolis induced apoptosis in the cells after 72 h of treatment and caused a dose-dependent increase in the cellular mRNA levels of p21CIP1 and p53. The results indicate that the Chinese propolis has antitumor effects and induces apoptosis in cell cultures of human colon carcinoma cells, strongly suggesting that the propolis contains components possessing antitumor activity [168]. The studies carried out with propolis from Asia showed that apoptosis by the mitochondrial pathway and the regulation of p53 could be the mechanism through which cell death is promoted by the extracts. Reports stated that some of the compounds commonly found in propolis are capable of regulating p53 [174]. This is promising for investigating the mechanisms of action of propolis in different types of cancer, so we think that it is necessary to replicate these studies using different cell lines and studies with 3D cultures. Recent advances in in vitro 3D culture technologies, such as organoids, have enabled the development of novel, more physiological human cancer models [175].

Regarding research conducted on the antitumor activity of propolis in North and South America, the antitumor effects have been investigated of propolis from southeastern and southern Brazil (at concentrations of 1–20 μg/mL) on metastasis-derived human prostate cancer cell lines (DU145 and PC-3) and a human prostate cancer-derived cell line (RC-58T/h/SA#4). Propolis showed clearer antitumor activity in DU145 and PC3. In addition, in the RC58T/h/SA#4 cells, propolis from southeastern Brazil induced antitumor activity that was associated with S phase arrest, and showed inhibition of cyclin D1, CDK4, and cyclin B1 expression, whereas propolis from southern Brazil induced antitumor activity that was associated with G2 arrest, and also showed higher induction of p21 expression. The results demonstrate that Brazilian propolis has an antitumor effect on human prostate cancer cells and promises to be a chemotherapeutic agent, in addition to a preventive agent, against prostate cancer [170]. The conclusions presented by the authors limited since the in vitro studies were partial and did not integrate cell interactions, heterogeneity, or the tumor microenvironment [176]. More mechanisms related to the regulation of the cell cycle most be described, as these works will provide the basis for continuing the study of the propolis in these regions in search of new alternatives for the treatment of cancer.

The study of cancer has also been investigated in different animal models. In this regard, antitumor properties of propolis from Iraq on HCT-116 in vivo have been evaluated. The experiments in HCT-116 tumor-bearing mice (athymic Fox N1-nu/nu mice) showed that oral administration of propolis (500 and 1000 mg/kg every day) for three weeks was associated with a decrease in mitotic cells, and increased p53 and decreased Ki-67 expression of cells in tumor sections, indicating that Iraqi propolis could be promising antitumor agent [166]. Another study investigated the protective effects of Iranian propolis (a supplemented propolis diet, 100 mg/mL) on N-methyl-N-nitro-N-nitrosoguanidine (MNNG)-induced gastric cancer in rats. It was observed that tumor incidence, the number of lesions, structural abnormalities, and beta-catenin of the group treated with propolis declined compared with the control. Propolis also induced the expression of proapoptotic Bax and reduced antiapoptotic Bcl-2 expression, suggesting that Iranian propolis could have a chemoprotective effect on MNNG-induced gastric cancer via inhibition of cell proliferation and apoptosis induction [169]. In 2019, Braga and collaborators evaluated Brazilian red propolis effects on colorectal preneoplastic lesions in rats induced by azoxymethane. After 16 weeks, treatment with propolis (100 mg/mL) reduced the number of aberrant crypt foci in the distal colon. In addition, propolis reduced the TBARS (thiobarbituric acid reactive substances) levels in peripheral blood, suggesting a protective activity of Brazilian propolis in the colon, possibly associated with its antioxidant properties [171]. Similarly, the effect of oral administration of Brazilian red propolis (10, 50 and 100 mg/kg) for 26 weeks on oral squamous cell carcinomas (OSCC) in rats induced by 9,10-dimethyl-1,2-benzanthracene (DMBA) was investigated. Propolis inhibited 40% of OSCC growth and promoted a three-week delay in development of clinically detectable tumors. Epithelial dysplasia was observed in all samples with no clinical tumor, except in the control suggesting, that Brazilian red propolis exerts chemopreventive effects on the progression of DMBA-induced epithelial dysplasia to OSCC in an experimental model of labial carcinogenesis [172]. The in vivo studies compiled in this section show promise in the search for new alternatives that complement cancer therapies. The evidence is still limited but strong effects have been reported, describing the cellular, biochemical, and molecular processes that provoke propolis extracts in animal models. Safety for human consumption must be ensured and effects in conjunction with cancer treatment must be verified [173,176].

Research has also shown that various phenolic compounds identified in propolis, such as flavonoids and other compounds, exhibit anti-cancer activity (Table 7) [177,178,179]. An in vitro study evaluated the antitumor activity of propolis from southeastern Poland on selected neoplastic cells. The propolis presented antitumor effects on human malignant melanoma cells (Me45) and HCT-116 cells in a dose-dependent manner (5–100 μg/mL), with HCT-116 cells being more susceptible to propolis. Me45 cell death occurred mainly through the apoptotic pathway, in contrast to HCT-116 cells, which after being exposed to propolis showed typical changes of cell necrosis. In addition, pinostrobin, kaempferol, galangin, chrysin, apigenin, quercetin, gallic acid, ferulic acid, caffeic acid, coumaric acid, and benzoic acid were identified, suggesting that the multi-directional interactions among the various chemical compounds in propolis appear to be the essential biological activities when considering its antitumor effects [180]. Similarly, the antitumor effects of a propolis (1–100 μg/mL) from southern Poland on nine human cancer cell lines (human colon cancer: Caco-2, DLD-1, and HT-29; human adenocarcinomic alveolar basal epithelial cells: H23 and A549; human breast cancer: MCF-7 and MDA-MB-468; human glioma: LN18 and U87) were analyzed. The propolis showed a strong antitumor effect on Caco-2 cells. The study also found decreased cell numbers in a dose-dependent manner in the H23, A549, MCF-7, MDA-MB-468, LN18, and U87 cells. Furthermore, various compounds, such as aromatic acids, fatty acids, flavonoids, chalcones, and some esters, were identified, indicating that propolis from Poland is a rich source of antitumoral compounds, and is thus a valuable natural product with the potential to improve human health [49]. Several studies have also reported that different samples of Brazilian propolis (red, green, and brown) from different regions (northeast, south, southeast, and mid-west of Brazil) showed antitumor effects on distinct cancer cell lines (B16F10 (murine melanoma), SF-295, OVCAR-8, HCT-116, HL-60, Jurkat, MOLT-4 (acute T cell leukemia), and K562 (chronic myelogenous leukemia)) at concentrations of 0.04–1000 μg/mL. Additionally, several compounds were identified in Brazilian propolis samples, such as coumaric acid, *p*-coumaric acid, gallic acid, caffeic acid, ferulic acid, cinnamic acid, artepillin C, genistein, kaempferol, catechin, epicatechin, daidzein, naringenin, pinobanksin, formononetin, pinostrobin, rutin, quercetin, luteolin, apigenin, terpenes, and tocopherol. Thus, the chemical composition and antitumor activity of Brazilian propolis probably varies depending on the geographical area of collection, and potentially presents a plethora of biologically active compounds that could have the potential to serve as an anticancer drug [34,181,182,183].

The antitumor activity of propolis and some of its identified pure compounds has also been studied (Table 8). A study evaluated the antitumor activity of a Polish propolis (6.25–200 μg/mL) and its derivative CAPE (2.5–80 μM) on two triple-negative breast cancer cell lines, MDA-MB-231 and Hs578T. Propolis and CAPE exhibited antitumor effects on MDA-MB-231 and Hs578T cells in a dose-dependent and exposure time-dependent manner. The results indicate that propolis and, in particular, CAPE, may markedly affect the viability of breast cancer cells, suggesting the potential role of bioactive compounds in chemoprevention or chemotherapy to potentiate the action of standard antitumor drugs [184]. Similarly, the Poland propolis (25–150 μg/mL), its individual components (chrysin, galangin, pinocembrin, caffeic acid, *p*-coumaric acid, ferulic acid at concentrations of 2.5–125 μg/mL), and their mixture (2.5–50 μg/mL for each one) exhibited antitumor effects on a human tongue squamous cell carcinoma cell line (CAL-27) in a dose dependent manner. Furthermore, the antitumor mechanism induced by these components was via apoptosis. The propolis activated caspases -3, -8, and -9, and the mixture of compounds was found to be the most potent inducer of apoptosis through both intrinsic and extrinsic pathways, suggesting that antitumor properties of propolis from Poland are related to the synergistic activities of its main components [185]. The studies referred to in this section demonstrate the importance of integrating the chemical composition of propolis. Although they are all in vitro studies, the effects of propolis on the different cancer cell lines can be better explained. We consider these anticancer properties to be due to compounds such as pinobanksin, chrysin, coumaric acid, and CAPE. Some studies are consistent with these works since they report the individual antitumor and antiangiogenic activities of these compounds [186,187,188,189,190]. This supports the use of propolis and its derivatives in more complex models for in vivo studies.

In 2019, Asgharpour and collaborators reported that Iranian propolis and its principal compound, quercetin, showed a dose-dependent antitumor effect (both at concentrations of 0–200 μg/mL) on A431 (skin squamous cell carcinoma) and KB (mouth epidermoid carcinoma) cell lines, indicating that the synergistic impact of the main components of Iranian propolis was related to the inhibition of cancer cell proliferation, and that it may also play a promising role in complementary medicine [191]. Similarly, anticancer effects of propolis from Saudi Arabia, and its bio-active ethyl acetate fraction (3.6 mg/mL), on Jurkat (acute T cell leukemia), A549, and HepG2 cells (hepatocellular carcinoma) were demonstrated. The major compounds in the propolis were triterpenoids, steroids, and diterpenoids. Propolis also presented antitumor effects on cancer cells in a concentration-dependent manner through apoptosis. Moreover, it was found that tubulin and/or microtubules are the cellular targets of the ethyl acetate fraction, demonstrating the importance of propolis from Saudi Arabia as an anticancer adjuvant candidate [54]. In another study, the effects of Chinese propolis (25, 50, and 100 μg/mL) and its major constituent (CAPE, 25 μg/mL) on inflammation-stimulated tumor and the Toll-like receptor 4 (TLR-4) signaling pathway, which plays a crucial role in human breast cancer cell line (MDA-MB-231), were evaluated. This treatment inhibited LPS-stimulated MDA-MB-231 cell line proliferation, migration, and nitric oxide production. Furthermore, propolis and CAPE activated caspase 3 and PARP to induce cell apoptosis, and also upregulated LC3-II and decreased p62 levels to induce autophagy during the process. TLR4 signaling pathway molecules, such as TLR4, MyD88, IRAK4, TRIF, and NF-κB p65, were all down-regulated after propolis and CAPE treatment in LPS-stimulated MDA-MB-231 cells. These results indicate that Chinese propolis and CAPE (its major constituent) inhibited breast cancer MDA-MB-231 cell proliferation in an inflammatory microenvironment via activating apoptosis and autophagy and inhibiting the TLR4 signaling pathway. Thus, Chinese propolis and CAPE may hold promising prospects in treating inflammation-induced tumor [192].

In other research, Egyptian propolis (0–100 μg/mL) and its combination with doxorubicin (0–100 mM) against the PC3 human prostate cancer cell line showed antitumor potential. Propolis activated cellular apoptosis and increased the mRNA expression levels for p53 and Bax genes, indicating that Egyptian propolis alone or in combination with doxorubicin showed greater antitumor and apoptotic effects against the PC3 cell line compared to doxorubicin treatment alone. Therefore, Egyptian propolis could be considered a promising candidate for prostate cancer chemotherapy [193]. This work is relevant since it provides an example of the path that can be followed for subsequent studies, since it combines a drug (doxorubicin) and propolis or its components to observe a possible synergistic effect against cancer cells. We consider this to be remarkable since some of the drugs currently used in the treatment of cancer, such as taxol, have an origin in alternative medicine, so we cannot rule out continuing to integrate compounds from herbal sources or beehives in cancer treatment [194].

In conclusion, both propolis and some of its bioactive compounds present significant therapeutic potential as cancer treatments because they may act through multiple pathways to reduce the development and other malignant characteristics of cancer cells. However, more research is still needed on the different forms of propolis that exist around the world to explain and understand the molecular mechanisms against cancer cells and obtain benefits for human health.

## 4. Conclusions

Traditional and alternative medicine is the world’s oldest form of health care and is used in the prevention and treatment of physical and mental illnesses in the present day. Propolis is a beekeeping product widely used in alternative medicine due to its easy accessibility. In this review, we compiled some of the biomedical properties of propolis, focusing mainly on diabetes, obesity, and cancer. Our search for information reflects the global trend to seek new alternatives for the treatment of these diseases. There is a significant volume of research on propolis that shows that it is able to help in the control of diabetes by lowering glucose levels, MDA, and pro-inflammatory cytokines, and assisting in weight loss. Furthermore, it enhances serum insulin levels, the translocation of GLUT-4, and the function of antioxidant enzymes, in addition to protecting cells and pancreatic function. In obesity, propolis has also shown benefits, with in vitro and animal models providing evidence that it induces the transcription of adiponectin and leptin, reduces the mass of visceral adipose tissue, and regulates the levels of triglycerides, non-esterified fatty acids, and cholesterol. Another relevant aspect for propolis is the data obtained on its antitumor effects related to the inhibition of the cell cycle, apoptosis, proliferation, viability, growth, and cell migration; it is notable that many of these activities continue to have selectivity towards tumor cells without affecting non-tumor cells. It is necessary to emphasize that the propolis of each geographical region has different biomedical activities due to the significant diversity that exists in its chemical composition. As a result, each geographical variant of propolis can be considered a great source of natural products, particularly terpenes and phenolic compounds, such as flavonoids. Each propolis is thus unique, requiring individual study. Despite all the attributes and virtues of propolis, several challenges remain to overcome. The first is to determine a classification in which all the properties that have been studied can be integrated as well as those that emerge from future investigations. The proposal by Bankova (2005) can function as basis for achieving adequate classification [6]. The standardization of propolis will lead to its safe and adequate consumption, to achieve this it is necessary to generate a commitment from the scientific community that works with this beekeeping product to describe the chemical composition of all the propolis extracts used in any research. Another challenge is to increase in vivo and clinical studies, since much of the available evidence of the biomedical properties of propolis is in vitro work, which, in many cases, prevents the reported activities from being directly applied in humans. Furthermore, we consider that future clinical studies should use propolis with a well-established chemical composition, since this will allow the establishment of a specific dose for each disease and the adequate treatment in infectious and non-infectious diseases. Investigation and better understanding of the properties of propolis, and phenomena such as synergism and other mechanisms of natural products, can assist in the development of new and better medicines and safe consumption treatments as complementary therapies for these diseases.

## Figures and Tables

**Table 1 nutrients-13-00078-t001:** Main compounds identified in propolis from different regions of the world.

Origin of Propolis	Compounds Identified	Ref.
Canada	Benzoic acid, cinnamic acid, vanillinic acid, *p*-coumaric acid, ferulic acid, caffeic acid, palmitic acid, oleic acid, pinocembrin, pinobanksin, chrysin, galangin, isosakuranetin, alpinone, kaempferol, pinostrobin chalcone, pinocembrin chalcone.	[27]
Mexico	Rutin, naringenin, hesperetin, pinocembrin, CAPE, chrysin, naringin, kaempferol, quercetin, acacetin, luteolin, pinostrobin, izalpinin, rhamnetin, galangin, alpinetin, dillenetin, cinnamic acid, caffeic acid, ferulic acid, syringic acid.	[28,29,30,31]
Brazil (red, green and brown)	Caffeic acid, gallic acid, transferulic acid, *p*-coumaric acid, catechin, epicatechin, kaempferol, formononetin, quercetin, rutin hydrate, luteolin, artepillin C, CAPE.	[32,33,34,35]
Southern Brazil	Gallic acid, caffeic acid and coumaric acid, artepillin C, pinocembrin.	[36]
Southern Nigeria	Preylated isoflavones, stilbenoid compounds.	[37]
Venezuela	Polyisoprenylated benzophenones and usual constituents found in samples of tropical regions.	[6,38,39]
Chile	Apigenin, pinocembrin, quercetin, CAPE.	[40]
Cameroon, Congo, Ethiopia and Kenya	Amyrin, lupeol, lupenona, ursolic acid, cycloartenol, ambonic acid and magniferolic acid.	[41,42,43]
Egypt	Ferulic acid, cis- and trans-caffeic acids, pinostrobin, galangine.	[44]
Morocco	Caffeic acid, *p*-coumaric acid, ferulic acid, naringenin, pinocembrin, chrysin, galangin, pinobanksin, and quercetin.	[45]
Argelia	Pinocembrin, chicoric and caffeic acids and their esters, galangin, diterpenic acids, chrysin, aromatic acids.	[46,47]
Greece	Protocatechuic acid, pinocembrin, kaempferol, apigenin, chrysin, galangin, chlorogenic acid, daidzein, ellagic acid, ferulicacid, gallicacid, hesperetin, hydroxytyrosol, luteolin, *p*-coumaric acid, pinobanksin, quercetin, tectochrysin, caffeic acid, sakuranetin, rhamnetin, CAPE, pinostrobin, syringic acid, kaempferide, acacetin, rutin, protocatechuic acid ethyl ester, resveratrol, phloridzin, maslinic acid, naringenin, eriodictyol, diosmetin, rosmarinic acid, myricetin, isorhamnetin, isosakuranetin, (+)-catechin, orientin, vitexin, trans-cinnamic acid, pinobanksin 3-O-acetate, cinnamylideneacetic acid, artepillin C, adipic acid, ursolic acid, suberic acid, genistein, hesperidin, corosolic acid, betulinic acid, isoferulic acid, naringin, tangeretin, diosmin, vanillin, chrysoeriol.	[48]
Poland	Benzoic acid, dihydrocinnamic acid, cinnamic acid, *p*-hydroxybenzoic acid, vanillic acid, *p*-coumaric acid, *p*-methoxycinnamic acid, ferulic acid, dimethoxycinnamic acid, isoferulic acid, caffeic acid, palmitic acid, linoleic acid, oleic acid, syringic acid, cichoric acid, pinostrobin chalcone, pinocembrin chalcone, pinocembrin, pinobanksin chalcone, sakuranetin, pinobanksin, chrysin, galangin, apigenin, kaempferol, quercetin, acacetin, (+/−)-pinostrobin, (+)-catechin, isosakuranetin chalcone, isosakuranetin, alpinon chalcone, alpinon and some esters.	[49,50]
China	Caffeic acid, *p*-coumaric acid, ferulic acid, iosuofmeraurlic acid, pinobanksin, naringenin, quercetin, kaempferol, apigenin, pinocembrin, chrysin, CAPE, galangin.	[51]
Turkey	Gallic acid, (±)-catechin, caffeic acid, syringic acid, epigallokatechin, *p*-coumaric acid, trans-ferulic acid, trans-isoferulic acid, myricetin, trans-cinnamic acid, daidzein, luteolin, pinobanksin, (±)-naringenin, apigenin, kaempferol, chrysin, pinocembrin, galangin, CAPE.	[52]

**Table 2 nutrients-13-00078-t002:** Propolis activity in different models of experimental diabetes.

Propolis Administered	Model/Diabetes Type	Effects	Ref.
Nigeria	Alloxan-induced Wistar rats	Decrease fasting blood glucose, reduces serum MDA, elevate serum SOD, improves histological in the pancreas and liver.	[89]
	Alloxan-induced rats	Decrease fasting blood glucose, VLDL and cholesterol, partially decrease HbA1c, increases HDL.	[90]
Morocco	STZ-induced type 2 diabetic Wistar rats	Lower blood glucose and lactic acid dehydrogenase, increases body weight, and ameliorates dyslipidemia and abnormal liver and kidney function caused by diabetes.	[93]
Saudi Arabia	STZ-induced type I diabetic mice	Improves body weight, decrease blood glucose, increases serum insulin, restore plasma cytokine (IL-1β, IL-6 and TNF-α), ROS levels and lipid profile to nearly normal levels.	[94]
	STZ-induced type 2 diabetic rats	Reduces glucose levels and lipid peroxide, increases SOD, CAT, and glutathione-S-transferase enzyme activities, ameliorates carboxymethyl lysine, IL-6 and immunoglobulins.	[79]
Malaysia	STZ-induced type 1 diabetic SD rats	Decrease the fasting blood glucose.	[95]
	STZ-induced type 2 diabetic SD rats	In both treatments, propolis and propolis combined with metformin, fasting blood glucose and serum insulin level decreased; body weight, pancreatic antioxidant enzymes, total antioxidant capacity, IL-10 and PCNA increased; MDA, NF-kB, TNF-α, IL-1β and cleaved caspase-3 decreased; histopathologically, islet area and number of beta cells increased comparable to normal control. Propolis from Malaysia showed in vitro α-glucosidase inhibition activity.	[96]
Iran	STZ-induced type 1 diabetic Wistar rats	Inhibits body weight loss, reduces serum glucose. Reduces MDA content, increases activity of SOD, GPx, and total antioxidant activity in the kidney tissue.	[97]
Indonesia	S961 peptide- induced type 2 diabetic mice (BALB/c)	Lower blood glucose level in a dose-dependent manner.	[98]
Taiwan	HFD-STZ-induced type 2 diabetic SD rats	Delay the development and progression of diabetes and reduces the severity of β-cell failure; attenuates the inflammation (serum TNF-α, IL-1β, and IL-6) and ROS; reduces MDA content; increases activity of SOD, GPx and total antioxidant activity in kidney tissue; promotes hepatic genes *PPAR-α* and *CYP7A1*.	[99]
Korea	Hepatocytes	Reduces expression and enzyme activity of G6Pase by inhibiting the phosphorylation of serine and tyrosine, which are involved in the activation of GSK3α and β, which maintain G6Pase gene expression.	[100]
China (encapsulated propolis)	HFD and low-dose STZ- induced type 2 diabetic SD rats	Attenuates fasting blood glucose and triglycerides, improves insulin action.	[101]
China and Brazil	STZ-induced type 1 diabetic SD rats	Chinese propolis: decrease HbA1c levels. Brazilian propolis: slightly decrease total cholesterol levels. Both propolis inhibits body weight loss, decrease blood glucose, reduces levels of alanine transaminase, aspartate transaminase, blood urea nitrogen and urine microalbuminuria-excretion rate.	[102]
Brazil (green propolis)	STZ-induced type 1 diabetic Wistar rats	Ameliorates body weight, decrease serum glucose, triglycerides, total cholesterol and LDL-C, increase serum HDL-C, reduces pancreatic MDA and serum NO, increases serum GSH and CAT, and pancreatic SOD.	[103]
Brazil	Non-insulin-dependent type 2 diabetic OLETF rats	Decrease plasma levels insulin and insulin resistance index HOMA-IR.	[104]
	OLETF rats	Decrease blood glucose associated with a reduction in plasma insulin levels.	[105]

**Table 3 nutrients-13-00078-t003:** Effect of propolis on diabetes and its main identified compounds.

Propolis Administered	Model/Diabetes Type	Identified Compounds	Effects	Ref.
China	STZ-induced type 1 diabetic rats	Caffeic acid, *p*-coumaric acid, ferulic acid, isoferulic acid, pinobanksin, quercetin, kaempferol, apigenin, pinocembrin, chrysin, CAPE, galangin.	Attenuates diabetes via directly decreasing the levels of fasting blood glucose and HbA1c; reduces MDA, ROS and RNS.	[112]
Mexico	STZ-induced type 1 diabetic mice	Pinocembrin, quercetin, naringin, naringenin, kaempferol, acacetin, luteolin, chrysin.	Decreases blood glucose and loss of body weight; increases plasma insulin levels and activities of the enzymes SOD, CAT, and GPx in serum.	[29]
Brazil	Skeletal muscle cells (L6 myotubes)	Artepillin C, coumaric acid, and kaempferide	Promotes GLUT-4 translocation and glucose uptake activity by PI3K and AMPK phosphorylation in a dose-dependent manner.	[110]

**Table 4 nutrients-13-00078-t004:** Studies on the effect of propolis administration in patients with type 2 diabetes.

Propolis Administered (Capsules)	Type Study/Characteristics of the Patients	Effects	Ref.
Iran	Randomized double-blind clinical trial. 94 type 2 diabetes mellitus patients. Propolis group: 55.40 ± 9.09 year; 33 female, 17 male. Placebo group: 54.86 ± 8.89 year; 28 female, 16 male.	Improves serum levels of HbA1c, two-hour postprandial glucose, blood insulin and insulin resistance indices (including HOMA-IR and HOMA-β); blood C-reactive protein and TNF-α levels decreases; notable reduction in serum liver transaminases (ALT and AST) and blood urea nitrogen concentrations.	[124]
	Randomized, double-blind, placebo-controlled, clinical trial. 60 type 2 diabetes mellitus patients. Propolis group: 51.81 ± 6.35 year; 30 patients, unspecified sex. Placebo group: 49.05 ± 8.2 year; 30 patients, unspecified sex.	Decreases fructosamine and the level of oxidized LDL, and CAT activity improves.	[125]
	Randomized, double-blind, placebo-controlled, clinical trial. 60 type 2 diabetes mellitus patients. Propolis group: 51.81 ± 6.35 year; 30 patients, female and male. Placebo group: 49.05 ± 8.2 year; 30 patients, female and male.	Decreases fasting blood glucose, two-hour postprandial glucose, blood insulin, insulin resistance index (HOMA-IR) and HbA1c; increases blood levels of total antioxidant capacity and activity of GPx and SOD.	[126]
	Randomized, double-blind, placebo-controlled, clinical trial. 57 type 2 diabetes mellitus patients. Propolis group: 51.3 ± 6.57 year; 17 female, 13 male. Placebo group: 56.07 ± 9.02 year; 11 female, 16 male.	Decreases fasting blood glucose, HbA1c, cholesterol total and LDL.	[127]
China	Randomized controlled trial. 61 type 2 diabetes mellitus patients. Propolis group: 57.7 ± 7.5 year; 20 female, 11 male. Control group: 60.6 ± 8.4 year; 16 female, 14 male.	Increases serum GSH and reduces LDH.	[128]
Brazil	Randomized controlled trial. 65 type 2 diabetes mellitus patients. Propolis group: 59.5 ± 8.0 year; 15 female, 18 male. Control group: 60.8 ± 8.6 year; 18 female, 14 male.	Increases GSH in blood and decreases LDH and TNF-α.	[129]

**Table 5 nutrients-13-00078-t005:** Efficacy of propolis in the regulation of obesity-related disorders.

Propolis Administered	In Vitro/In Vivo Model	Effects	Ref.
Brazil (red propolis)	3T3-L1 preadipocytes	Induces adiponectin mRNA, probably through activation of the adiponectin promoter by PPAR-γ.	[142]
Brazil (green propolis)	Differentiated 3T3-L1 adipocytes	Elevates leptin expression.	[143]
	HFD-induced obese C57BL/6 mice	Suppress feeding and increases leptin mRNA production in the visceral adipose tissues.	
	C57BL/6JHamSlc-ob/ob mice	Decreases mesenteric adipose tissue mass, plasma cholesterol levels attenuate.	[146]
Brazil	HFD-induced obese C57BL/6N mice	Reduces body weight gain, weight of visceral adipose tissue, liver and serum triglycerides, cholesterol, and non-esterified fatty acids; down-regulates of mRNA expression associated with fatty acid biosynthesis, including fatty acid synthase, acetyl-CoA carboxylase a, and sterol regulatory element binding protein in liver.	[147]
	HFD-induced C57BL/6 mice	Decreases blood triglyceride levels	[148]
	HFD-induced obese Wistar rats	Repress weight gain of mesenteric, perirenal, and total white adipose tissues; liver PPAR-α protein level was higher; liver HMG-CoA reductase protein and level of protein PPAR-γ in the adipose tissues were lower; reduces level of cholesterol and triglyceride in plasma and liver.	[149]
China	HFD-induced C57BL/6 mice	Reduces body weight and improves insulin resistance in a dose-dependent manner; reverses liver weight loss and triglyceride accumulation in association with hepatic steatosis.	[150]

**Table 6 nutrients-13-00078-t006:** Antitumor activity of propolis reported in in vitro and in vivo models.

Propolis Administered	Patient Cells/ Cell Lines/Model	Effects	Ref.
Poland	SCC-25.	Affects cell viability and cellular proliferation; presents cancer-selective antiproliferative activity.	[162]
Portugal (Northeast and Centre)	Cancerous renal cells derived from renal carcinoma patients; A-498.	Both exhibits selective antiproliferative activity against malignant cells; inhibits cell growth of cancerous renal cells derived from renal carcinoma patients in a concentration-dependent manner. In A-498 cell line, propolis from northeast of Portugal exerted higher antitumor activity than propolis from central Portugal.	[163]
Portugal	HCT-15.	Exhibits antitumor activity both in terms of inhibition of viability and cell death in a dose- and time-dependent way; decreases glucose consumption and lactate production.	[164]
	MDA-MB-468, MCF7, 22RV1, U251, SW1088.	Decreases cell viability of tumor cells; decreases MDA-MB-231 and DU145 cell proliferation and migration, with cell cycle changes, and increased cell death.	[165]
Iraq	HL-60, HCT-116.	Induces antitumor effect in HL-60 cells, the apoptosis was associated with down-regulation of Bcl-2 and activation of Bax, and the increased levels of γ-H2AX in a dose dependent manner; induces inhibitory effect in colony potential of HCT-116 cells and necrotic features were observed, size of cells was dramatically increased by swelling of cytoplasm and loss of membrane integrity, cell rupture and release of cellular contents; induces cell cycle perturbations in both cell lines.	[166]
	HCT-116 tumor-bearing mice	Administration of propolis was associated with a decrease in mitotic cells, increased p53 and decreased Ki-67 expression of cells in tumor sections.	
Korea	HL-60	Decreases proliferation dose- and time-dependent manner; induces activation of caspase-3 and ADP-ribose polymerase; induces release of cytochrome c from mitochondria to cytosol.	[167]
China	HCT-116, HT29, SW480.	Causes dose-dependent antitumor activity in all cells. In HCT-116, induces apoptosis and causes a dose-dependent increase in the cellular mRNA levels of p21CIP1 and p53.	[168]
Thailand	A549, HeLa.	Presents antitumor activity.	[161]
Iran	MNNG-induced gastric cancer rats.	Declines tumor incidence, number of lesions, structural abnormalities, and beta-catenin; induces the expression of proapoptotic Bax and reduces antiapoptotic Bcl-2 expression.	[169]
Mexico	HeLa, SiHa, CaSki.	Presents antitumor activity.	[31]
Brazil	OVCAR-8, HCT-116, SF-295.	Presents antitumor activity.	[56]
Brazil (southeastern and southern)	DU145, PC-3, RC-58T/h/SA#4.	Both presents antitumor activity more marked in DU145 and PC3; in RC58T/h/SA#4 cells, propolis from southeastern induces antitumor activity that was associated with S phase arrest, and showed inhibition of cyclin D1, CDK4 and cyclin B1 expression; propolis from southern induces antitumor activity that was associated with G2 arrest, and also showed higher induction of p21 expression.	[170]
Brazilian (red propolis)	Azoxymethane-induced colorectal preneoplastic lesions in rats.	Reduces number of aberrant crypt foci in distal colon and TBARS levels in peripheral blood.	[171]
	DMBA-induced oral squamous cell carcinomas in rats.	Inhibits 40% of oral squamous cell carcinomas growth and promoted a three-week delay in development of clinically detectable tumors.	[172]

**Table 7 nutrients-13-00078-t007:** Antitumor properties of propolis and identification of its chemical composition.

Propolis Administered	Cell Lines	Identified Compounds	Effects	Ref.
Poland	Me45, HCT-116	Pinostrobin, kaempferol, galangin, chrysin, apigenin, quercetin, gallic acid, ferulic acid, caffeic acid, coumaric acid and benzoic acid.	Presents antitumor activity in a dose-dependent manner; HCT-116 cells are more susceptible and showed typical changes for cell necrosis; Me45 cells death was mainly through apoptotic pathway.	[180]
South Poland	Caco-2, DLD-1, HT-29, H23, A549, MCF-7, MDA-MB-468, LN18, U87.	Aromatic acids, fatty acids, flavonoids, chalcones and some esters.	Strong antitumor effect on Caco-2 cells; decreases cell numbers in a dose-dependent manner in cells H23, A549, MCF-7, MDA-MB-468, LN18 and U87.	[49]
Brazilian (red, green and brown propolis from Northeast, South, Southeast and Midwest)	B16F10, SF-295, OVCAR-8, HCT-116, HL-60, Jurkat, MOLT-4, K562	Coumaric acid, *p*-coumaric acid, gallic acid, caffeic acid, ferulic acid, cinnamic acid, artepillin C, genistein, kaempferol, catechin, epicatechin, daidzein, naringenin, pinobanksin, formononetin, pinostrobin, rutin, quercetin, luteolin, apigenin, terpenes, tocopherol.	Presents antitumor activity.	[34,181,182,183]

**Table 8 nutrients-13-00078-t008:** Different propolis and their main bioactive compounds with antitumor effect.

Propolis/Compound Administered	Cell Lines	Effects	Ref.
Poland/CAPE	MDA-MB-231, Hs578T.	Both presents antitumor activity in a dose-dependent and exposure time-dependent manner.	[184]
Poland/chrysin, galangin, pinocembrin, caffeic acid, *p*-coumaric acid, ferulic acid	CAL-27.	Propolis, individual components and mixture of compounds presents antitumor activity in a dose dependent manner; antitumor mechanism induced by these components was through apoptosis. The propolis activates caspases -3, -8, -9, and mixture of compounds are the most potent inducer of apoptosis thorough both intrinsic and extrinsic pathway.	[185]
Iran/quercetin	A431, KB.	Both showed a dose-dependent antitumor effect.	[191]
Saudi Arabia/ethyl acetate fraction	Jurkat, A549, HepG2.	Propolis presents antitumor activity in a concentration-dependent manner through apoptosis; tubulin and/or microtubules are the cellular targets of the ethyl acetate fraction.	[54]
China/CAPE	MDA-MB-231	Both activates caspase 3 and PARP to induce cell apoptosis, upregulated LC3-II and decreases p62 level to induces autophagy during the process, down-regulates molecules TLR4, MyD88, IRAK4, TRIF and NF-κB p65; propolis inhibits proliferation, migration and nitric oxide production.	[192]
Egypt	PC3	Activates cellular apoptosis and increased the mRNA expression levels for p53 and Bax genes.	[193]

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
