# Peer review of "Biomedical Properties of Propolis on Diverse Chronic Diseases and Its Potential Applications and Health Benefits"

_nutrients, 2020, doi:10.3390/nu13010078_

Round 1

Reviewer 1 Report

In the current manuscript, Rivera-Yañez et al. provided a thorough summary of the biomedical properties and applications of propolis. The authors first compared the chemical compositions of propolis with different geographic origins. They then reviewed the studies focusing on the potential beneficial effects of propolis on the treatment of chronic diseases including diabetes, obesity, and cancers. This paper provided a high-level overview of the chemical properties and potential clinical applications of propolis, however, there are some issues that need to be addressed. 

1) The chemical compositions vary in geographically different propolis. But how this may affect the biomedical proproperty? What are the bioactivities of the main components, and how they determined the treatment potential of propolis? The authors cited Siheri,W et al's review paper, but it is not sufficient and more background and original research studies should be discussed in detail.

2) The effects of propolis on diseases: The current manuscript is to some extent a list of conclusions from a larger number of studies, which are easily accessible to readers. Therefore, a good reveiw paper should provide more. What are the limitations of these studies? Are the in vitro or in vivo models reliable? Are the conclusions scientifically sound?  If not, what kind of studies are needed to overcome these limitations?

3) In the dissucstion section, again it is important to cover the controversies in the field, and the problems to be solved in the near future.

Author Response

Reviewer 1's corrections:

1) The chemical compositions vary in geographically different propolis. But how this may affect the biomedical proproperty? What are the bioactivities of the main components, and how they determined the treatment potential of propolis? The authors cited Siheri,W et al's review paper, but it is not sufficient and more background and original research studies should be discussed in detail.

 Regarding the comment, we add:

Line 163

, bees collect tree buds, saps, and resins from different plant sources to produce various types of propolis, that is to say, the chemical composition of propolis is mainly dependent on the plant species in the area [52, 54-56]. These variations in its compostion directly influences its biomedical properties. There are even reports that showing that propolis extract from the same country, for example Brazil or China, have variations of the antioxidant, antimicrobial, antiparasitic, cytotoxic and antitumoral properties, this is due to its great geographic extension and the great diversity of its flora [34, 57, 58]. These chemical variations determine the biomedical properties and in some cases the nutraceutical characteristics of each propolis. Although, for many researchers, this characteristic of propolis seems a negative point for its study and application, our work group considers that it is a virtue since thanks to this we naturally have a different product with unique properties in each geographical region of each country, which can be a great source of new therapeutic alternatives thanks to the innumerable interactions, such as synergism and antagonism between the great diversity of secondary metabolites that make up propolis[59-61]. For this reason, we also consider that it is of great importance to continue with the study of propolis whose biological properties and chemical composition have not been determined.

On the other hand, it has recently been proposed that several abiotic factors such as light intensity, average temperature, humidity, wind speed, solar radiation, water availability, and rainfall, collectively defined as seasonal effects, can directly affect the concentration of plant components, including flavonoids and phenolic acids, which influence with the chemical composition and bioactivity of propolis [62-64]. This aspect difficult propolis classification and standardization, and should be taken into account that different solvents and extraction method, may influencing its activity [32, 65]. In this way, a universal standardization would be impossible, however, it has been proposed that propolis biological properties should be linked to a detailed investigation of its chemical composition and to its botanical sources[6, 19]. A classification has been proposed to group propolis into seven types: poplar, birch, green (alecrim), red (Clusia), Pacific and Canarian [6, 19]. This classification it is proposed to group propolis into seven types: poplar, birch, green (alecrim), red (Clusia), Pacific and Canarian [6]. Some works support this categorization, for example, these show that Macaranga-type Pacific propolis from different countries have a very similar chemical composition and biomedical activities [64, 66-69]. However, more research is still needed to achieve a reliable classification, specifying the characteristics of propolis from regions with arid or semi-arid climatic conditions of Africa and North America (like Mexico) where the vegetation has a large number of endemic species. Furthermore, a limitation of this classification is due to the fact that it only focuses on the identification of bioactive compounds that meet any of the seven proposed profiles, and does not fully consider minority compounds and the synergistic or antagonistic effects they exert on biological activities. Standardization is urgent for the correct application of propolis in the food and pharmaceutical industries in the near future [70, 71]. Therefore, in this review we will address the different studies specifying the country of origin of each propolis sample.

2) The effects of propolis on diseases: The current manuscript is to some extent a list of conclusions from a larger number of studies, which are easily accessible to readers. Therefore, a good reveiw paper should provide more. What are the limitations of these studies? Are the in vitro or in vivo models reliable? Are the conclusions scientifically sound?  If not, what kind of studies are needed to overcome these limitations?

Regarding the comment, we add:

Line 198

Although studies on diabetes animals models induced with alloxan are not the most reliable in terms of the similarity of the damage process of this disease, they can give us a first approach to know if propolis shows any activity on diabetes, however, it is necessary corroborate the results with a model that better resembles the course of diabetes, such as streptozotocin-induced animal models. We agree with the reported of Jain et al. (2011) and Misra et al. (2012) that alloxan is a doubtful drug as regards the induction of experimental diabetes. This, however, remains contradictory to many studies where alloxan has been successfully used for the same [92, 93]. Another great limitation of these works is the absence of the chemical composition, and as we have already mentioned, this is essential for a correct application of propolis in the pharmaceutical and medical area [71].

Line 232

Research studying the effect of propolis in type I diabetes shows different activities such as reduction of glucose and HbA1c levels, improvement of the lipid profile, restoration of antioxidant capacity, and an inhibition of body weight loss. The results are consistent and suggest an antidiabetic effect, however, many authors agree, and we also, that it is necessary to carry out studies that evaluate more effects pharmacological and biochemical of propolis, focused on the search for new, better and more efficient complementary therapies for the treatment of diabetes [107, 108].

Line 277

As mentioned in all previous research studying the effect of propolis in animal models of type 2 diabetes, this natural product has a wide variety of antidiabetic properties, since it can control blood glucose, modulate lipid metabolism, improve insulin sensitivity, increase the activity of antioxidant enzymes in different tissues, an increase in the area of the islets and the number of beta cells is even mentioned by some authors, however, we consider that the conclusions provided by the authors are still limited, so that need to implement different studies focused on elucidating the mechanisms of action by which propolis has an effect on diabetes [109, 110].

Line 299

We consider that these effects are due to what was reported by some authors that have administered flavonoids, such as quercetin, naringin, or genistein, and have informed reduced blood glucose concentrations and detected insulin in serum or islets, these effects likely resulted from changes in Ca2+ metabolism, thus facilitating the hypoglycaemic effects of flavonoids [114-117]. Moreover, different studies have reported that the administration of some flavonoids, such as acacetin, luteolin, chrysin, kaempferol, or naringenin, decreases glucose levels in diabetic rats and mice, and  inhibitory effects against the enzymatic activity of ?-glucosidase [118-121]. These results are promising, so in the future it would be important to take it to clinical studies administering propolis or its main bioactive compounds, in fact some clinical studies are already focused on the application of compounds such as quercetin in the treatment of diabetes [122].

Line 311

As mentioned above, there is a wide variety of studies evaluating the effect of propolis and some of its bioactive components on diabetes, however, although this is promising, it is necessary to evaluate these possible mechanisms of action in in vivo models [123], since they have to take into account other factors such as oxidative stress, chronic low-grade inflammation, and the response of each individual to treatment, which would help to better understand the disease and the possible complementary treatment for it [123, 124].

Line 354

Although there are few clinical studies that mention some properties of propolis on diabetes, most of the research is with Iranian propolis, for these reason, the clinical studies with propolis from different regions should be done, since that it is important to consider the variations in the biomedical properties of different propolis. In addition, it should be noted that the biggest omission in these clinical studies is that they do not report the chemical composition of propolis and neither do they mention whether there is a history of in vitro and in vivo studies that supports whether these propolis have any type of biological activity or how it was decided to implement clinical studies[129, 130].

Line 383

Although in vitro studies are of great help to give us an overview of the activity of propolis on obesity, most evaluate only a chemical or molecular aspect, so its effect is still partial and the studies do not take into account that this disease is multifactorial, therefore, in vivo studies are always essential to reach clinical trials [145, 146].

Line 426

In the in vivo studies mentioned in this review, the different effects of propolis on obesity are reported through various mechanisms suppressing feeding in mice, inducing leptin in adipocytes, reducing adipose tissue mass, and improving dyslipidemia. However, we consider that this is a disease that encompasses a large number of factors that need to be taken into account for the study of obesity, which continues to be a challenge in the field of research, which is why there are few studies, since in order to mention an optimal effect of propolis on this disease, is essential to do a large investigation that can combine aspects related to obesity such as diabetes, metabolic syndrome, hypertension, just to mention a few, to reach a conclusion more certain about the activity of propolis on obesity, in addition to the fact that clinical studies are also necessary about this complex disease [153].

Line 437

Exist different studies that report that in growing preadipocytes, quercetin extensively decreases the expression of LPL, sterol regulatory element-binding protein 1c (SREBP1c), and PPAR-γ, a key adipogenic transcription factor [156, 157]. Quercetin also caused dose- and time-dependent increases in lipolysis in rat adipocytes [158].The anti-obesity effect of naringenin is through the reduction in adipose tissue mass and inhibition of preadipocyte proliferation [159]. Moreover, this compound also remarkably increased the activity of various enzymes required for fatty acid oxidation in hepatocytes, such as acetyl-coenzyme A acetyltransferases (ACAT, also known as thiolase), acyl-coenzyme A oxidase, carnitine O-octanoyl transferase (COT, also known as medium-chain/long-chain carnitine acyltransferase), and 3-ketoacyl-coenzyme A [160].

Line 461

It is necessary to clarify that these studies report the evaluation of various biomedical activities of different extracts of propolis, so the reported evidence is limited to a cytotoxic approach, therefore we consider that it is necessary to include controls with non-malignant cells to determine if the effect is selective for cancer cells. However, only one of the reports the chemical composition and thanks to this, it has a solid basis for discussion. The description of the chemical composition is of great relevance for some authors since they mention that without it the systematic classification of propolis will be impossible [6].

Line 493

Although in vivo tests are not carried out in these works, they do have very relevant results since the antitumor, antiproliferative and antiangiogenic effect of several European propolis extracts is demonstrated. So we consider that the next step is to carry out more in vivo studies to consolidate the use of propolis and its components, and in this way to reach development of clinical studies. In fact, rodent models of cancer have revolutionized our ability to study gene and protein functions in vivo and to better understand their molecular pathways and mechanisms [168].

Line 516

The studies carried out with propolis from Asia show that apoptosis by the mitochondrial pathway and the regulation of p53 could be the mechanism of cell death promoted by the extracts of propolis, exist reports that some of the compounds commonly found in propolis are capable of regulating p53 [176]. This is promising to investigate the mechanisms of action of propolis in different types of cancer, so we believe that it is necessary to replicate these studies using different cell lines and studies with 3D cultures. Recent advances in in vitro 3D culture technologies, such as organoids, have generated the development of novel, more physiological human cancer models [177].

Line 527

The conclusions presented by the authors are limited since the in vitro studies are very partial and do not integrate cell interactions, heterogeneity and the tumor microenvironment [178]. It is still necessary to describe more mechanisms related to the regulation of the cell cycle, due these works are the basis for continuing to study the propolis of these regions in search of new alternatives for the treatment of cancer.

Line 550

The in vivo studies compiled in this section are really promising in the search for new alternatives that complement cancer therapies, the evidence is still few but strong effects are reported, the description of the cellular, biochemical and molecular processes that provoke propolis extracts in animal models used. These preclinical studies have to overcome the challenge of being safe for human consumption and effects to complement cancer treatment [168, 178].

Line 582

The studies referred to in this section will demonstrate the importance of integrating the chemical composition since, although they are all in vitro studies, the effects of propolis on the different cancer cell lines can be better explained. We consider that these anticancer properties are due to compounds such as pinobanksin, chrysin, coumaric acid y CAPE. There are studies that are consistent with these works since they report the individual antitumor and atiangiogenic activity of these compounds [185-189]. This supports the use of propolis and its derivatives in more complex models for in vivo studies.

Line 625

This work is of great relevance since it is an example of the path that can be followed for subsequent studies, since it combines a drug (doxorubicin) and propolis or its components, to observe a possible synergistic effect against cancer cells. We consider this to be remarkable since some of the drugs currently used in the treatment of cancer, such as taxol, have an origin in alternative medicine, so we cannot rule out continuing to integrate compounds from herbal sources or beehives in the cancer treatment [196].

3) In the dissucstion section, again it is important to cover the controversies in the field, and the problems to be solved in the near future.

Regarding the comment, we add:

Line 493

Although in vivo tests are not carried out in these works, they do have very relevant results since the antitumor, antiproliferative and antiangiogenic effect of several European propolis extracts is demonstrated. So we consider that the next step is to carry out more in vivo studies to consolidate the use of propolis and its components, and in this way to reach development of clinical studies. In fact, rodent models of cancer have revolutionized our ability to study gene and protein functions in vivo and to better understand their molecular pathways and mechanisms [168].

Line 516

This is promising to investigate the mechanisms of action of propolis in different types of cancer, so we believe that it is necessary to replicate these studies using different cell lines and studies with 3D cultures. Recent advances in in vitro 3D culture technologies, such as organoids, have generated the development of novel, more physiological human cancer models [177].

Line 550

The in vivo studies compiled in this section are really promising in the search for new alternatives that complement cancer therapies, the evidence is still few but strong effects are reported, the description of the cellular, biochemical and molecular processes that provoke propolis extracts in animal models used. These preclinical studies have to overcome the challenge of being safe for human consumption and effects to complement cancer treatment

Line 650

Despite all the attributes and virtues of propolis, there are still several challenges to overcome. The first is to determine a classification in which all the properties that have been studied can be integrated and those that emerge in a new investigation. The proposal by Bankova (2005) can function as basis for achieving an adequate classification [6]. The standardization of propolis will lead to a safe and adequate consumption of it, to achieve this it is necessary to generate a commitment from the scientific community that works with this beekeeping product to inform the chemical composition of all the propolis extract used in any research. Another challenge is to increase in vivo and clinical studies, since much of the evidence we have of the biomedical properties of propolis is in vitro work that in many cases it is difficult for the reported activities to be directly applicable in humans. Furthermore, we consider that future clinical studies should use propolis with a well-established chemical composition, since this will allow to establish a specific dose for each disease and an adequate treatment in infectious and non-infectious diseases.

Note: We understand that corrections are large and if you consider that a grammar revision is required we accept it.

Reviewer 2 Report

  1. This review article collect a number of propolis studies around the world, it seems to be interested by this scientific filed. Propolis has a large number of biological activities, it has been extensively studied by the world's researchers since 1990s.
  2. The chemical composition and biological activity of propolis vary greatly based on the different plants that bees may collect. Unfortunately, the authors do not target the category of propolis worldwide, which leads to a very vague reader's understanding of propolis. In addition, the authors only describe the names of the ingredients contained in the propolis produced in various places, do not indicate the content of these important ingredients, and may make the readers feel un-focused.
  3. Based on the plant source and geographical location, propolis is categorized into seven groups (Sforcin and Bankova, 2011, Journal of Ethnopharmacology 133: 253–260). Of these, Pacific propolis, found in the Pacific islands, originates from Macaranga spp., is known as Macaranga-type Pacific propolis. Authors seem not mention about this Macaranga-type Pacific propolis.
  4. Macaranga-type propolis are rich in prenylated flavonoids and are reported to have a wide range of pharmacological benefits, including antioxidant, anti-inflammation, anticancer, antidiabetic and longevity-extending effects (Shahinozzaman et al., 2020, Phytotherapy Research 2020;1–16). It is worth to discuss in this manuscript.
  5. This manuscript focus on the anti-diabetes, obesity and cancer activities of different propolis, but authors didn’t present the dosage effects in vitro and/or in vivo studies.
  6. Line 180-181: …. diabetes was estimated to be 463 million in 2019, and is 180 projected to reach 463 million by 2030 and 700 million by 2045…., this paragraph seems to have a mistake.

Author Response

Reviewer 2's corrections:

  1. This review article collect a number of propolis studies around the world, it seems to be interested by this scientific filed. Propolis has a large number of biological activities, it has been extensively studied by the world's researchers since 1990s.

Regarding the comment, we add:

Indeed this is the objective of the work and the corrections helped to give a better focus of the review.

  1. The chemical composition and biological activity of propolis vary greatly based on the different plants that bees may collect. Unfortunately, the authors do not target the category of propolis worldwide, which leads to a very vague reader's understanding of propolis. In addition, the authors only describe the names of the ingredients contained in the propolis produced in various places, do not indicate the content of these important ingredients, and may make the readers feel un-focused.

Regarding the comment, we add:

Line 163

bees collect tree buds, saps, and resins from different plant sources to produce various types of propolis, that is to say, the chemical composition of propolis is mainly dependent on the plant species in the area [52, 54-56]. These variations in its compostion directly influences its biomedical properties. There are even reports that showing that propolis extract from the same country, for example Brazil or China, have variations of the antioxidant, antimicrobial, antiparasitic, cytotoxic and antitumoral properties, this is due to its great geographic extension and the great diversity of its flora [34, 57, 58]. These chemical variations determine the biomedical properties and in some cases the nutraceutical characteristics of each propolis. Although, for many researchers, this characteristic of propolis seems a negative point for its study and application, our work group considers that it is a virtue since thanks to this we naturally have a different product with unique properties in each geographical region of each country, which can be a great source of new therapeutic alternatives thanks to the innumerable interactions, such as synergism and antagonism between the great diversity of secondary metabolites that make up propolis[59-61]. For this reason, we also consider that it is of great importance to continue with the study of propolis whose biological properties and chemical composition have not been determined.

On the other hand, it has recently been proposed that several abiotic factors such as light intensity, average temperature, humidity, wind speed, solar radiation, water availability, and rainfall, collectively defined as seasonal effects, can directly affect the concentration of plant components, including flavonoids and phenolic acids, which influence with the chemical composition and bioactivity of propolis [62-64]. This aspect difficult propolis classification and standardization, and should be taken into account that different solvents and extraction method, may influencing its activity [32, 65]. In this way, a universal standardization would be impossible, however, it has been proposed that propolis biological properties should be linked to a detailed investigation of its chemical composition and to its botanical sources[6, 19]. A classification has been proposed to group propolis into seven types: poplar, birch, green (alecrim), red (Clusia), Pacific and Canarian [6, 19]. This classification it is proposed to group propolis into seven types: poplar, birch, green (alecrim), red (Clusia), Pacific and Canarian [6].

Regarding “In addition, the authors only describe the names of the ingredients contained in the propolis produced in various places, do not indicate the content of these important ingredients, and may make the readers feel un-focused.” We add:

Line 168

Propolis has been extensively investigated, because its constituents exhibit several properties that can be applied to treat these diseases. Most of the propolis present bioactivities such as antioxidant and anti-inflammatory capacity, effects on the regulation of the cell cycle and antitumor [49, 57, 74-76]. These properties are due to the individual components of propolis, which are highly variable (table 1) and are mainly grouped into phenolic acids or their esters, flavonoids, terpens, aromatic aldehydes and alcohols, fatty acids, stilbens, and ?-steroids [21, 22].

  1. Based on the plant source and geographical location, propolis is categorized into seven groups (Sforcin and Bankova, 2011, Journal of Ethnopharmacology 133: 253–260). Of these, Pacific propolis, found in the Pacific islands, originates from Macaranga spp., is known as Macaranga-type Pacific propolis. Authors seem not mention about this Macaranga-type Pacific propolis.

Regarding the comment, we add:

Line 163, second paragraph

A classification has been proposed to group propolis into seven types: poplar, birch, green (alecrim), red (Clusia), Pacific and Canarian [6, 19]. This classification it is proposed to group propolis into seven types: poplar, birch, green (alecrim), red (Clusia), Pacific and Canarian [6]. Some works support this categorization, for example, these show that Macaranga-type Pacific propolis from different countries have a very similar chemical composition and biomedical activities [64, 66-69]. However, more research is still needed to achieve a reliable classification, specifying the characteristics of propolis from regions with arid or semi-arid climatic conditions of Africa and North America (like Mexico) where the vegetation has a large number of endemic species. Furthermore, a limitation of this classification is due to the fact that it only focuses on the identification of bioactive compounds that meet any of the seven proposed profiles, and does not fully consider minority compounds and the synergistic or antagonistic effects they exert on biological activities. Standardization is urgent for the correct application of propolis in the food and pharmaceutical industries in the near future [70, 71]. Therefore, in this review we will address the different studies specifying the country of origin of each propolis sample.

  1. Macaranga-type propolis are rich in prenylated flavonoids and are reported to have a wide range of pharmacological benefits, including antioxidant, anti-inflammation, anticancer, antidiabetic and longevity-extending effects (Shahinozzaman et al., 2020, Phytotherapy Research 2020;1–16). It is worth to discuss in this manuscript.

Regarding the comment, we add:

Line 168

A large number of studies of propolis from different countries have generated vast evidence of the beneficial effects of this bee product. Several reports place a particular emphasis on the varios activities of Macaranga-type Pacific propolis like antioxidant, anti-inflammation, anticancer, antidiabetic and longevity-extending effects, besides, its chemical composition is one of the best described [64, 67, 68, 78, 79]. In this revision we will compile works carried out with propolis of different origins and we will integrate the studies carried out with Macaranga-type propolis but referring to the country of origin.

  1. This manuscript focus on the anti-diabetes, obesity and cancer activities of different propolis, but authors didn’t present the dosage effects in vitro and/or in vivo studies.

Regarding the comment, we add:

The doses used in all the studies were included in the text.

  1. Line 180-181: …. diabetes was estimated to be 463 million in 2019, and is 180 projected to reach 463 million by 2030 and 700 million by 2045…., this paragraph seems to have a mistake.

Regarding the comment, we add:

Line 180

 According to Saeedi et al. the International Diabetes Federation reported that the prevalence of diabetes was estimated to be 463 million in 2019, and is projected to reach 463 million by 2030 and 700 million by 2045 [89].

Note: We understand that corrections are large and if you consider that a grammar revision is required we accept it.

Round 2

Reviewer 1 Report

The authors have addressed my concerns. It should be ready for publication after the grammar check.

Author Response

Reviewer's 1 suggestion

The authors have addressed my concerns. It should be ready for publication after the grammar check.

Reply:

The manuscript was sent a grammar revision in the service offered by MDPI, the English edition ID is: English-25410.
